# Fluid: **A Unified Evaluation Framework for Flexible Sequential Data**

**Matthew Wallingford**                                      *mcw244@cs.washington.edu*
*University of Washington*

**Aditya Kusupati**                                       *kusupati@cs.washington.edu*
*University of Washington*

**Keivan Alizadeh**                                        *keivanalizadeh@gmail.com*
*University of Washington*

**Aaron Walsman**                                           *aaronwalsman@gmail.com*
*University of Washington*

**Aniruddha Kembhavi**                                              *anik@allenai.org*
*Allen Institute for Artificial Intelligence*

**Ali Farhadi**                                                    *ali@cs.uw.edu*
*University of Washington*

**Reviewed on OpenReview:** *https://openreview.net/forum?id=UvJBKWaSSH*

## Abstract

Modern machine learning methods excel when training data is large-scale, well labeled, and matches the test distribution. Learning in less ideal conditions remains an open challenge. The sub-fields of few-shot, continual, transfer, and representation learning have made substantial strides in learning under adverse conditions, each affording distinct advantages through methods and insights. These methods address different challenges such as data arriving sequentially or scarce training examples, however often the difficult conditions an ML system will face over its lifetime cannot be anticipated prior to deployment. Therefore, general ML systems which can handle the many challenges of learning in practical settings are needed. To foster research towards the goal of general ML methods, we introduce a new unified evaluation framework – Fluid (Flexible Sequential Data). Fluid integrates the objectives of few-shot, continual, transfer, and representation learning while enabling comparison and integration of techniques across these subfields. In Fluid, a learner faces a stream of data and must make sequential predictions while choosing how to update itself, adapt quickly to novel classes, and deal with changing data distributions; while accounting for the total amount of compute. We conduct experiments on a broad set of methods which shed new insight on the advantages and limitations of current techniques and indicate new research problems to solve. As a starting point towards more general methods, we present two new baselines which outperform other evaluated methods on Fluid.

## 1 Introduction

Modern ML methods have demonstrated remarkable capabilities, particularly in settings with large-scale labeled training data drawn IID. However, in practice the learning conditions are often not so ideal. Consider a general recognition system, a key component in many computer vision applications. One would expect such a system to learn from new data distributions, recognize classes with few and many examples, revise the set of known classes as novel ones are seen, and update itself over time using new data.

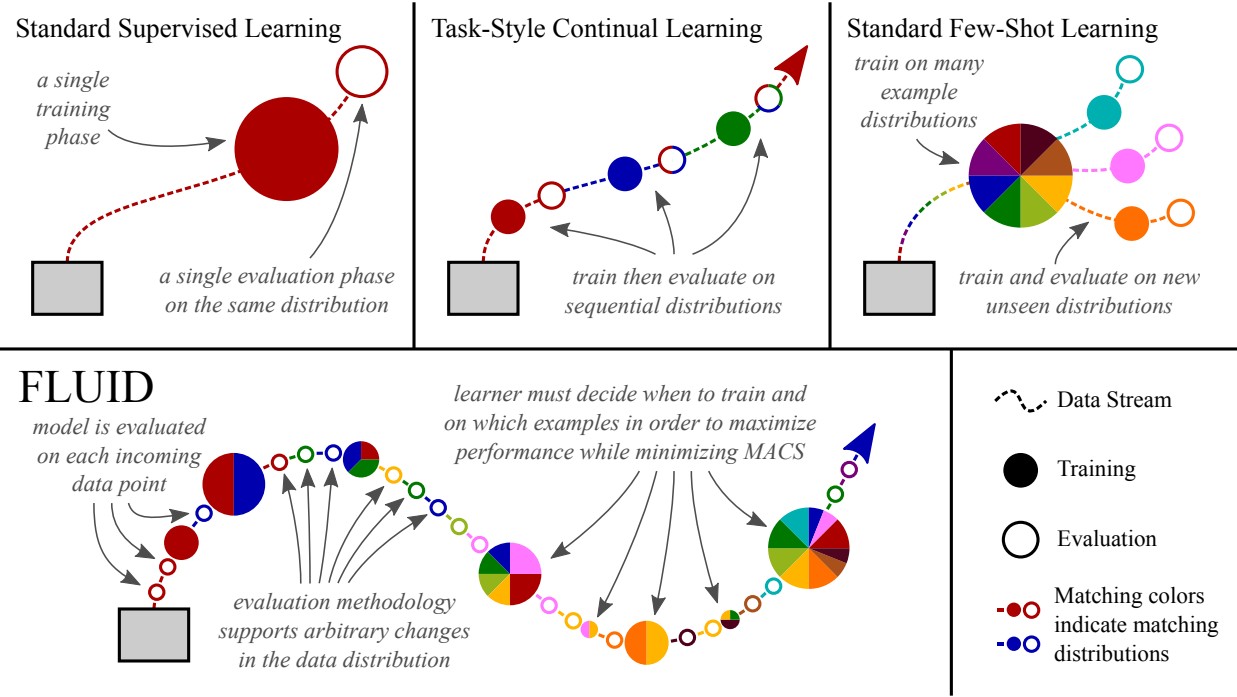

Figure 1: Comparison of supervised (top-left), continual (top-middle), and few-shot learning (top-right) with FLUID (bottom). The learner (grey box) accumulates data (dotted path), trains on given data (filled nodes), then is evaluated (empty nodes). The size of the node indicates the scale of the training or evaluation. Each color represents a different set of classes.

Various subfields such as few-shot, continual, transfer, and representation learning have made substantial progress on the challenges associated with learning in non-ideal settings. The methods from these fields excel when the deployment conditions can be anticipated and align with specific scenarios. For example, few-shot methods perform well when the number of new classes and examples per class are few and known in advance. Similarly, continual learning techniques improve performance when data from new distributions arrive in fixed-size batches at predictable intervals. However, in many applications the exact conditions cannot be known a priori and are likely to change over time. For example, computer vision systems for autonomous self-checkout systems must account for changing inventories, distribution shifts in background and lighting between stores, and a long-tailed distribution of products, among other real-world challenges (Polacco & Backes, 2018; Wankhede et al., 2018). This calls for general methods that can handle a plethora of scenarios during deployment.

To make progress towards such general methods, new evaluations which reflect the key aspects of learning in practical settings are essential. But, *what are these aspects?* We posit the following as some of the necessary elements: (1) *Sequential Data* - In many application domains the data streams in. ML methods must be capable of learning from sequential data and new distributions. (2) *X-shot* - Data often has a different number of examples for each class (few for some and many for others). Current evaluations assume prior knowledge of which data regime (few-shot, many-shot, etc.) new classes will be from, but often this cannot be known in advance. (3) *Flexible Training Phases* – Practical scenarios rarely delineate when and how a system should train. ML systems should be capable of making decisions such as when to train based on incoming data, which data to train with, and whether to update all parameters or just the classifier. (4) *Compute Aware* – Real-world systems often have computational constraints not only for inference but also for training. ML systems should account for the total compute used throughout their lifetime. (5) *Open-world* - As new data is encountered the set of known classes may change over time. Learning in practical settings often entails recognizing known classes while detecting when data comes from new classes.

With these elements in consideration we introduce FLUID (**f**lexible seq**u**ent**i**al **d**ata), a unified evaluation framework. FLUID integrates the objectives of few-shot, continual, transfer, and representation learning into a simple and realizable formulation along with a benchmarkable implementation. In FLUID, a learner is deployed on a stream of data from an unknown distribution and must classify incoming samples one at a time while deciding when and how to update based on newly received data.

We conduct extensive experiments with FLUID on a broad set of methods across subfields. The experimental results quantitatively demonstrate the current limitations and capabilities of various ML approaches. For example, we find that current few-shot methods do not scale well to more classes and a varying number of examples. Similarly, we observe that catastrophic forgetting is a significant challenge in the FLUID setting which is not solved by existing continual learning approaches. Finally, we briefly investigate the unexplored problem of update strategies for deciding when and how to train efficient on incoming data. The framework, data and models will be open-sourced.

**We make the following contributions:**

1. We propose a new evaluation framework, FLUID, which unifies the objectives of few-shot, continual, transfer, and self-supervised learning into a simple benchmark that enables comparison and integration of methods across related subfields and presents new research challenges.

2. We present empirical findings from experiments with FLUID which demonstrate the utility of more general evaluations. Specifically, we find that existing few-shot methods do not scale well to the FLUID setting which has more classes and varying number of examples per class. Larger networks perform better for few-shot classes, contrary to prevailing thought in few-shot works which train light-weight models to avoid overfitting (Snell et al., 2017; Finn et al., 2017; Sun et al., 2019; Xu et al., 2020). We find this discrepancy to be caused by meta-training decreasing performance for larger networks whereas supervised pretraining does not. We observe in the FLUID setting that freezing network parameters prevents catastrophic forgetting and learns novel classes better than existing continual learning methods and suggests significant room for improvement.

3. We introduce two baselines, Exemplar Tuning & Minimum Distance Thresholding, which outperform evaluated methods in FLUID while matching performance in supervised and few-shot settings.

## 2 Related Work

We discuss the key aspects of FLUID in the context of other works and evaluations. We compare existing frameworks in Table 1 and discuss FLUID in the context of real-world applications in appendix R.

**Sequential Data and Continual Learning** New data is an inevitable consequence of our dynamic world and learning over time is a long-standing challenge (Thrun, 1996). In recent years, continual learning (CL) has made notable progress on the problem of learning in a sequential fashion (Li & Hoiem, 2017; Kirkpatrick et al., 2017; Rebuffi et al., 2017; Aljundi et al., 2018; 2019; Riemer et al., 2019). Several setups have been proposed in order to evaluate systems' abilities to learn continuously and primarily focus on *catastrophic forgetting*, a phenomenon where models drastically lose accuracy on old tasks when trained on new tasks. The typical CL setup sequentially presents data from each task then evaluates on current and previous tasks (Li & Hoiem, 2017; Kirkpatrick et al., 2017; Rebuffi et al., 2017). Recent variants have proposed a task-free setting where the data distribution changes without the model's knowledge (Harrison et al., 2019; Riemer et al., 2019; He et al., 2020a; Wortsman et al., 2020; Sun et al., 2020).

Recently several works have empirically investigated the disconnect between existing continual learning evaluations and real-world applications (Prabhu et al., 2020; Hussain et al., 2021; Onl, 2022). Such works show that implicit details in the experimental setups can lead to significantly different methods and empirical conclusions. For example, Prabhu et al. (2020) show that not accounting for compute allows for an unrealistic method which retrains from scratch to drastically outperform sophisticated CL methods. In FLUID, we aim to address unrealistic assumptions and carefully consider each detail of the experimental design.

There are two assumptions in existing CL evaluations which we remove in FLUID. The first assumption is that data will be received in large batches with ample data for every class in the task. This circumvents a

Table 1: We categorize existing evaluations frameworks aimed at learning in practical settings. ✓: presence; ✗: absence & −: not applicable.

| Property / Framework | Open-world | Sequential | Variable Batch-size | Few-shot | Many-shot | Compute Aware | Memory Constrained | Flexible Training | Distribution Shift | Non-Stationary |
|---|---|---|---|---|---|---|---|---|---|---|
| Representation Learning | ✗ | ✗ | – | ✗ | ✓ | – | – | – | ✓ | – |
| Transfer Learning | ✗ | ✗ | – | ✗ | ✓ | – | – | – | ✓ | – |
| Task-based Continual Learning | ✗ | ✓ | ✗ | ✗ | ✓ | – | ✓ | ✗ | ✓ | ✓ |
| Task-free Continual Learning | ✗ | ✓ | ✗ | ✗ | ✓ | – | ✓ | ✗ | ✓ | ✓ |
| Few-shot Learning | ✗ | ✗ | – | ✓ | ✗ | – | – | – | ✓ | – |
| Generalized Few-shot Learning | ✗ | ✗ | – | ✓ | ✓ | – | – | – | ✓ | – |
| Streaming Perception | ✗ | ✓ | – | – | – | ✓ | – | – | ✗ | ✗ |
| Open Long-Tailed Recognition | ✓ | ✗ | – | ✓ | ✓ | – | – | – | ✗ | ✗ |
| Data Stream Classification | – | ✓ | ✓ | – | ✓ | – | ✓ | – | ✓ | – |
| Test Time Training | ✗ | ✓ | ✓ | – | – | – | – | ✗ | ✓ | ✗ |
| OSAKA | ✓ | ✓ | ✓ | ✓ | ✓ | – | – | ✓ | ✓ | ✓ |
| FLUID (Ours) | ✓ | ✓ | ✓ | ✓ | ✓ | ✓ | – | ✓ | ✓ | ✗ |

fundamental challenge of sequential data which is learning new classes from only a few examples. Consider the common scenario in which a learner encounters an instance from a novel class. The system must determine that it belongs to a new class with no previous examples (zero-shot learning and out-of-distribution detection). The next time an instance from the category appears, the system must be capable of one-shot learning, and so forth. In other words, few-shot learning is an emergent requirement of learning from sequential data. The second assumption is that the training and testing phases will be delineated to the system. Deciding when to train and which data to train on is an intrinsic challenge of learning continuously.

Some CL evaluations include a memory cache to store images, typically between 0 - 1000 examples from previous tasks. We argue that setting a specific memory constraint, particularly this small, is too constraining. Methods should account for memory, but the FLUID framework does not explicitly restrict memory during streaming. Note that all methods use memory caching during the sequential phase except for the Nearest Class Mean (NCM) baseline.

Data stream classification (Gomes et al., 2019; Wankhade et al.; Stefanowski & Brzezinski, 2017; Bifet et al., 2010) has worked on the problem of learning from sequential data. This line of work primarily focuses on traditional classification and not image recognition. At a high level FLUID has similar goals as data stream classification. FLUID differs in its implementation while integrating the preexisting fields of few-shot, transfer, representation, and continual learning.

Most closely related to FLUID is the OSAKA benchmark (Caccia et al., 2020). Caccia et al. (2020) propose a more general scenario which unifies meta-learning, meta-continual learning, and continual-meta learning and addresses limitations of the previous evaluations. FLUID builds upon this direction of research with a few key differences. First, FLUID accounts for compute consumed throughout training, which is important metric to consider when flexible training phases are allowed. Second, FLUID samples from a long-tail distribution and evaluates accuracy with respect to class frequency (head and tail class accuracy). This allows us to study the trade-off between meta-learning methods which excel in the few-shot regime and methods which are better with large-scale data. Last, we conduct the experiments at the ImageNet scale. We find that the larger-scale setting leads to new empirical findings such as meta-training not scaling to larger networks and more data.

One key distinction that OSAKA and other CL frameworks incorporate that is not included in FLUID is a non-stationary distribution. In general, FLUID differs from traditional continual learning formulations (van de Ven et al., 2022; De Lange et al., 2021) in that the goal is to perform well on the deployment distribution, rather than current and previously seen distributions. In FLUID the system adapts to one unknown distribution shift, whereas traditional CL frameworks change distributions over multiple episodes.

**Few-shot and X-shot Learning** Learning from few examples for some classes is an inherent aspect of the real-world. Learning from large, uniform datasets (Russakovsky et al., 2015; Lin et al., 2014) has been the primary focus of supervised learning while few-shot learning has gained traction as a subfield (Ravi & Larochelle, 2017; Hariharan & Girshick, 2017; Oreshkin et al., 2018; Sun et al., 2019).

While few-shot learning is a step towards more generally applicable ML methods, the framework has assumptions that are unlikely to hold in practical settings. The experimental setup for few-shot is typically

the $n$-shot $k$-way evaluation. Models are trained on base classes during *meta-training* and then tested on novel classes during *meta-testing*. The $n$-shot $k$-way experimental setup is limited in two respects. $n$-shot $k$-way assumes that a model will always be given exactly $n$ examples for $k$ classes at test time which is unrealistic. Second, most works only evaluate 5-way scenarios with 1, 5, and 10 shots. Realistic settings often have a mix of classes from both the high and low data regime. Recently, more general variants of the few-shot benchmark have been proposed which introduce variable shot numbers and greater domain difference between datasets (Chao et al., 2016; Triantafillou et al., 2020; Dumoulin et al., 2021).

Fluid naturally integrates the few-shot problem into its framework by sequentially presenting data from a long tail distribution and evaluates systems across a spectrum of shots and ways. Our experimental results on canonical few-shot methods indicate that methods are overly tuned to the specific conditions of the few-shot evaluation which indicates the need for more general experimental frameworks such as Fluid.

**Flexible Training Phases**  Current experimental setups dictate when models will be trained and tested. Ideally, an ML system should decide when to train itself, what data to train on, and what to optimize for (Cho et al., 2013). By removing the assumption that training and testing phases are fixed and known in advance, Fluid provides a benchmark for tackling the unexplored challenge of learning when to train.

**Compute Aware**  ML systems capable of adapting to their environment over time must account for the computational costs of their learning strategies as well as of inference. Prabhu et al. (Prabhu et al., 2020) showed that current CL frameworks do not measure total compute and therefore a naive, but compute-hungry strategy can drastically outperform state of the art methods. Previous works have focused on efficient inference (Rastegari et al., 2016; Howard et al., 2017; Kusupati et al., 2020; 2021; 2022; Lin et al., 2021; Wallingford et al., 2022) and some on training costs (Evci et al., 2020). In Fluid we measure the total compute for both learning and inference over the sequence.

**Open-world**  Practical scenarios entail inferring in an open world - where the classes and number of classes are unknown to the learner. Few-shot, continual, and traditional supervised learning setups assume that test samples can only come from known classes. Previous works explored static open-world recognition (Liu et al., 2019; Bendale & Boult, 2015; Kong & Ramanan, 2021; Vaze et al., 2021; Radford et al., 2021) and the related problem of out-of-distribution detection (Hendrycks & Gimpel, 2016; Masana et al., 2018; Lee). Fluid is a natural integration of sequential and open-world learning where the learner must identify new classes and update its known class set throughout the stream.

## 3  Fluid **Evaluation Details**

Fluid evaluation is designed to be simple and general while integrating the key aspects outlined in section 2.

Table 2: The evaluation metrics used in the Fluid framework to capture the performance and capabilities of various algorithms.

| Metric | Description |
|---|---|
| Overall Accuracy | Accuracy over the sequence. |
| Mean Per-Class Accuracy | Accuracy averaged over all classes in the sequence. |
| Total Compute | MAC operations for all compute over the sequence. |
| Unseen Class Detection | AUROC for the detection of OOD samples. |
| Cross-Sectional Accuracies | Classes in the sequence belong fall into 4 subcategories: 1) *Pretraining-Head*: $> 50$ samples & in pretraining. 2) *Pretraining-Tail*: $\leq 50$ samples & in pretraining. 3) *Novel-Head*: $> 50$ samples & not in pretraining. 4) *Novel-Tail*: $\leq 50$ samples & not in pretraining. |

**Formulation** Let learning system $S$ be composed of a model, $f_\theta : \mathbf{x} \mapsto \mathbf{y}$, and update strategy, $U :$ $f_\theta \times \bigcup_{t=1}^{T} (x_t, y_t) \mapsto f_{\theta'}$, where $\bigcup_{t=1}^{T} (x_t, y_t)$ is the training data collected up to time $T$. Model, $f_\theta$, may be initialized using pretraining data $D = \{x_i, y_i\}_{i=1}^{n}$.

At each new time step, $t+1$, the model is given a sample, $x_t$, and provides a class label, $f_\theta(x_{t+1}) \in \{1...K+1\}$, for $K$ known classes. In other words, the sample may belong to one of $K$ previously seen classes, or a new class. The model output is evaluated with respect to the true label, $\mathbb{1}\{f_\theta(x_{t+1}) = y_{t+1}\}$, and $(x_{t+1}, y_{t+1})$ is added to the training set. If $y_{t+1}$ is from a new class, the set of known classes is updated accordingly. Next the model, $f_\theta$, may be updated according to $U$ using all previously observed data. This process is repeated for some total number of time steps.

Systems are evaluated on a suite of metrics including the overall and mean class accuracy throughout the stream along with the total compute required for updates and inference.

**Data** In this paper, we evaluate methods with FLUID using a subset of ImageNet-22K (Deng et al., 2009). Traditionally, few-shot learning used datasets like Omniglot (Lake et al., 2011) & MiniImagenet (Vinyals et al., 2016) and continual learning focused on MNIST (LeCun, 1998) & CIFAR (Krizhevsky et al., 2009). Some recent continual learning works have used Split-ImageNet (Wen et al., 2020). The aforementioned datasets are mostly small-scale and have very few classes. We evaluate on the ImageNet-22K dataset to present new challenges to existing models. Recently, the INaturalist (Van Horn et al., 2018; Wertheimer & Hariharan, 2019) and LVIS (Gupta et al., 2019) datasets have advocated for heavy-tailed distributions. We follow suit and draw our sequences from a heavy-tailed distribution.

The dataset consists of a pretraining dataset and 5 different sequences of images for streaming (3 test and 2 validation sequences). For pretraining we use the standard ImageNet-1K (Russakovsky et al., 2015). This allows us to leverage existing models built by the community as pre-trained checkpoints. Sequence images come from ImageNet-22K after removing ImageNet-1K's images. Each test sequence contains images from 1000 different classes, 750 of which do not appear in ImageNet-1K. We refer to the overlapping 250 classes as Pretrain classes and the remaining 750 as Novel classes. Each sequence is constructed by randomly sampling images from a heavy-tailed distribution of these 1000 classes. Each sequence contains $\sim 90000$ samples, where head classes contain $> 50$ and tail classes contain $\leq 50$ samples. The sequence allows us to study how methods perform on combinations of pretrain vs novel, and head vs tail classes. In Table 3, we show results obtained for sequence 5, and the Appendix I shows results across all test sequences. More comprehensive statistics on the data and sequences are in Appendix B.

**Pretraining** Supervised pretraining (He et al., 2016) on large annotated datasets like ImageNet facilitates the transfer of learnt representations to help data-scarce downstream tasks. Unsupervised learning methods like autoencoders (Tschannen et al., 2018) and more recent self-supervised methods (Jing & Tian, 2020; Purushwalkam & Gupta, 2020; Gordon et al., 2020) like Momentum Contrast (MoCo) (He et al., 2020b) and SimCLR (Chen et al., 2020a) have begun to produce representations as rich as that of supervised learning and achieve similar accuracy on various downstream tasks.

Before the sequential phase, we pretrain our model on ImageNet-1K. In our experiments, we compare how different pretraining strategies (contrastive learning, meta-training, & supervised training) perform under more adverse conditions. We find new insights such as contrastive representations perform significantly worse on few-shot classes compared to their supervised counterparts in the FLUID evaluation.

**Evaluation metrics** Table 2 defines the evaluation metrics in FLUID to gauge the performance of the algorithms.

## 4  Baselines and Methods

In this section, we summarize the baselines, other methods, and our proposed baselines, Exemplar Tuning and MDT. Additional details about the methods and implementation can be found in Appendix D and Appendix E respectively.

**Standard Training and Fine-Tuning** We evaluate standard model training (update all parameters in the network) and fine-tuning (update only the final linear classifier) with offline batch training. We ablate over the number of layers trained during fine-tuning in Appendix F.

**Nearest Class Mean (NCM)** Recently, multiple works (Tian et al., 2020; Wang et al., 2019) have found that Nearest Class Mean (NCM) is comparable to state-of-the-art few-shot methods (Sun et al., 2019; Oreshkin et al., 2018). NCM in the context of deep learning performs a 1-nearest neighbor search in feature space with the centroid of each class as a neighbor. We pretrain a neural network with a linear classifier using softmax cross-entropy loss, then freeze the parameters to obtain features.

**Few-shot Methods** We evaluate the following methods: MAML (Finn et al., 2017), Prototypical Networks (PTN) (Snell et al., 2017), Weight Imprinting (Qi et al., 2018), ProtoMAML (Triantafillou et al., 2020), SimpleCNAPS (Bateni et al., 2020), New Meta-Baseline (Chen et al., 2020b) and ConstellationNet (Xu et al., 2020).

PTN trains a deep feature embedding using 1-nearest neighbor with class centroids and soft nearest neighbor loss. Parameters are trained with meta-training and backprop.

MAML is a gradient-based approach which uses second-order optimization to learn parameters that can be quickly adapted to a given task. We tailor MAML to FLUID by pretraining the model according to the objective in Appendix D and then fine-tune during the sequential phase.

Weight Imprinting initializes the weights of a cosine classifier as the class centroids, then fine-tunes with a learnable temperature. For further analysis of Weight Imprinting and comparison to Exemplar Tuning see Appendix M.

Meta-Baseline is same in implementation as NCM except a phase of meta-training is done after standard batch training.

For further details on the remaining the methods and how they are implemented in FLUID see Appendix D.

**Continual Learning (CL) Methods** We evaluate Learning without Forgetting (LwF) (Li & Hoiem, 2017), Elastic Weight Consolidation (EWC) (Kirkpatrick et al., 2017), Dark Experience Replay(DER) (Buzzega et al., 2020), & Experience Replay with Asymmetric Cross-Entropy (ER-ACE) (Caccia et al., 2022) to observe whether continual learning techniques improve performance in FLUID.

LwF leverages knowledge distillation (Buciluă et al., 2006) to retain accuracy on previous training data without storing it. EWC enables CL in a supervised learning context by penalizing the total distance moved by the parameters from the optimal model of previous tasks weighted by the corresponding Fisher information. Unlike LwF, EWC requires stored data, typically the validation set, from the previous tasks. In FLUID, we use LwF and EWC to retain performance on pretrain classes. For the continual learning methods we train all network parameters according to standard training procedures (Appendix E). For further details on ER-ACE and DER see Appendix D.

**Out-of-Distribution (OOD) Methods** We evaluate two methods proposed by Hendrycks & Gimpel (Hendrycks & Gimpel, 2016) (HG) and OLTR (Liu et al., 2019) along with our proposed OOD baseline. The HG baseline thresholds the maximum probability output of the softmax classifier to determine whether a sample is OOD.

We propose the baseline, **Minimum Distance Thresholding (MDT)**, which utilizes the minimum distance from the sample to all class representations, $c_i$. In the case of NCM the class representation is the class mean and for a linear layer it is the $i$th column vector. For distance function $d$ and a threshold $\tau$, a sample is out of distribution if: $\mathbf{I}(\min_i d(c_i, \mathbf{x}) < t)$. MDT with a Nearest Class Mean classifier can be derived from a Dirichlet process mixture model (Hjort et al., 2010), where a sample is considered to be out of distribution if it is assigned to a new cluster. The concentration parameter for the Chinese Restaurant process can be related to the out of distribution threshold $\tau$ as:

$$\tau = 2\sigma \log \left( \frac{\alpha}{\left(1 + \frac{\rho}{\sigma}\right)^{d/2}} \right)$$

$\rho$ is the covariance scaling of the gaussian prior over the cluster means, $\boldsymbol{\mu} \sim \mathcal{N}(\mathbf{0}, \rho I)$ and $\sigma$ is the scaling for the isotropic covariance of each gaussian cluster, $\mathcal{N}(\boldsymbol{\mu}_{\boldsymbol{z}_i}, \sigma I)$. Similar to the DP-Means derivation, as $\sigma$ goes to 0, the probability of a sample x being assigned to a new cluster goes to 1 when the distance of x to the closest cluster exceeds $\tau$.

Other metric learning techniques have proposed using distance to detect out of distribution examples (Lee; Masana et al., 2018). MDT primarily differs from these works in that it can be used with a standard classification network and can be performed in a single forward pass with negligible extra compute.

**Exemplar Tuning (ET)** We present a new baseline that leverages the inductive biases of instance-based methods and parametric deep learning. The traditional classification layer is effective when given a large number of examples but performs poorly when only a few examples are present. On the other hand, NCM and other few-shot methods are accurate in the low data regime but do not significantly improve when more data is added. Exemplar Tuning (ET) synthesizes these methods in order to initialize class representations accurately when learning new classes and to have the capacity to improve when presented with more data. We formulate each class representation (classifier), $C_i$, and class probability as the following:

$$C_i = \frac{1}{n} \sum_{x \in D_i} \frac{f(x; \theta)}{\|f(x; \theta)\|} + \mathbf{r}_i; \quad p(y = i|x) = \frac{e^{C_i \cdot f(x; \theta)}}{\sum_{i \neq j} e^{C_j \cdot f(x; \theta)}} \tag{1}$$

where $f(x; \theta)$ is a parametrized neural network, $\mathbf{r}_i$ is a learnable residual, $n$ is the number of class examples, and $D_i$ are all examples in class $i$. $C_i$ is analogous to the $i$-th column vector in a linear classification layer.

The class centroid (the first term of $C_i$ in Eq 1) provides an accurate initialization from which the residual term $\mathbf{r}_i$ can continue to learn. Thus ET is accurate for classes with few examples (where deep parametric models are inaccurate) and continues to improve for classes with more examples (where few-shot methods are lacking). In our experiments, the centroid is updated after each sample for minimal compute and batch train the residual vector with cross-entropy loss according to the same schedule as fine-tuning (see Appendix E for details).

We compare ET to initializing a cosine classifier with class centroids and fine-tuning (Weight Imprinting). Exemplar Tuning outperforms Weight Imprinting and affords two significant advantages besides better accuracy. 1) ET has two frequencies of updates (fast instance-based and slow gradient-based) which allows the method to quickly adapt to distribution shifts while providing the capacity to improve over a long time horizon. 2) ET automatically balances between few-shot & many-shot performance, unlike Weight Imprinting which requires apriori knowledge of when to switch from centroid-based classification to fine-tuning.

## 5 Experiments and Analysis

We evaluate an array of methods from few-shot, continual, self-supervised learning, and out-of-distribution detection in the proposed FLUID framework. We present a broad set of empirical findings which validate the need for more general evaluations such as FLUID and suggest future research directions. Table 3 displays a comprehensive set of metrics for the set of methods (outlined in Sec 4). Throughout this section, we will refer to rows of the table for specific analysis. For a summary of the main insights see Sec 5.7.

### 5.1 Few-shot Analysis

We evaluate three groups of methods to understand the effects of meta-training on generalization to novel classes and to gauge the overall utility of current few-shot methods in FLUID. The first group consists of methods which are purely meta-trained (Prototypical Networks and MAML). The second group consists of methods that do not utilize meta-training (Weight Imprinting, NCM, and fine-tuning). Finally, we evaluate methods which use both meta-training and standard batch training (ProtoMAML, SimpleCNAPS, ConstellationNet, Meta-Baseline).

We observe the methods that purely meta-train (PTN and MAML) do not perform well in the large-scale FLUID setting with over 30% lower overall accuracy than the NCM baseline (Table 3 and Figure 2-a). One might argue that PTN and MAML could simply scale to the larger setting by increasing model capacity. However,

Table 3: Performance of the suite of methods (outlined in Section 4). We present several accuracy metrics - Overall, Mean-per-class as well as accuracy bucketed into 4 categories: Novel-Head, Novel-Tail, Pretrain-Head and Pretrain-Tail (Pretrain refers to classes present in the ImageNet-1K dataset).

| | Method | Pretrain Strategy | Novel - Head (>50) | Pretrain - Head (>50) | Novel - Tail (<50) | Pretrain - Tail (<50) | **Mean Per-Class** | **Overall** | GMACs↓ ($\times 10^6$) |
|---|---|---|---|---|---|---|---|---|---|
| | | | Backbone - Conv-4 | | | | | | |
| FSL | (a) Prototypical Networks | Meta | 11.63 | 22.03 | 6.90 | 13.26 | 11.13 | 15.98 | **0.06** |
| | (b) MAML | Meta | 2.86 | 2.02 | 0.15 | 0.10 | 1.10 | 3.64 | 2.20 |
| | | | Backbone - ResNet18 | | | | | | |
| FSL | (c) Prototypical Networks | Meta | 8.64 | 16.98 | 6.79 | 12.74 | 9.50 | 11.14 | 0.15 |
| | (d) ConstellationNet | Sup+Meta. | 39.26 | 65.35 | 26.28 | 52.91 | 40.21 | 46.13 | 0.16 |
| | (e) Meta-Baseline | Sup.+Meta | 40.47 | 67.03 | 27.53 | 53.87 | 40.23 | 47.62 | 0.16 / 5.73 |
| | (f) ProtoMAML | Sup.+Meta | 41.68 | 69.48 | 28.91 | 54.16 | 42.94 | 49.25 | 5.73 |
| | (g) SimpleCNAPS | Sup.+Meta | 41.59 | 68.79 | 25.79 | 52.16 | 41.82 | 48.23 | 0.16 |
| | (h) Weight Imprinting | Sup. | 40.32 | 67.46 | 15.35 | 34.18 | 32.69 | 48.51 | 0.16 / 5.73 |
| | (i) OLTR | Sup. | 40.83 | 40.00 | 17.27 | 13.85 | 27.77 | 45.06 | 0.16 / 6.39 |
| CL | (j) LwF | Sup. | 30.07 | 67.50 | 7.23 | 56.96 | 31.02 | 48.76 | 22.58 / 45.16 |
| | (k) EWC | Sup. | 39.03 | 70.84 | 16.59 | 47.18 | 34.89 | 50.39 | 12.29 |
| | (l) DER | Sup. | 35.34 | 74.41 | 10.64 | 52.05 | 32.30 | 49.07 | 11.29 |
| | (m) ER-ACE | Sup. | 33.13 | 69.61 | 16.59 | 49.00 | 31.34 | 44.50 | 12.29 |
| Baselines | (n) Fine-tune | Sup. | 43.41 | **77.29** | 23.56 | **58.77** | 41.54 | 53.80 | 0.16 / 5.73 |
| | (o) Standard Training | Sup. | 38.51 | 68.14 | 16.90 | 43.25 | 33.99 | 49.46 | 11.29 |
| | (p) NCM | Sup. | 42.35 | 72.69 | **31.72** | 56.17 | 43.44 | 50.62 | **0.15** |
| | (q) Exemplar Tuning **(Ours)** | Sup. | **48.85** | 75.70 | 27.93 | 45.73 | **43.61** | **58.16** | 0.16 / 5.73 |
| MoCo | (r) Weight Imprinting | MoCo | 16.77 | 26.98 | 6.19 | 8.69 | 12.60 | 22.90 | 0.16 / 5.73 |
| | (s) OLTR | MoCo | 34.60 | 33.74 | 13.38 | 9.38 | 22.68 | 39.92 | 0.16 / 6.39 |
| | (t) Fine-tune | MoCo | 14.49 | 27.59 | 0.10 | 4.96 | 8.91 | 26.86 | 0.16 / 5.73 |
| | (u) Standard Training | MoCo | 26.63 | 45.02 | 9.63 | 20.54 | 21.12 | 35.60 | 11.29 |
| | (v) NCM | MoCo | 19.24 | 31.12 | 14.40 | 21.95 | 18.99 | 22.90 | 0.15 |
| | (w) Exemplar Tuning **(Ours)** | MoCo | 31.50 | 46.21 | 12.90 | 21.10 | 24.36 | 39.61 | 0.16 / 5.73 |

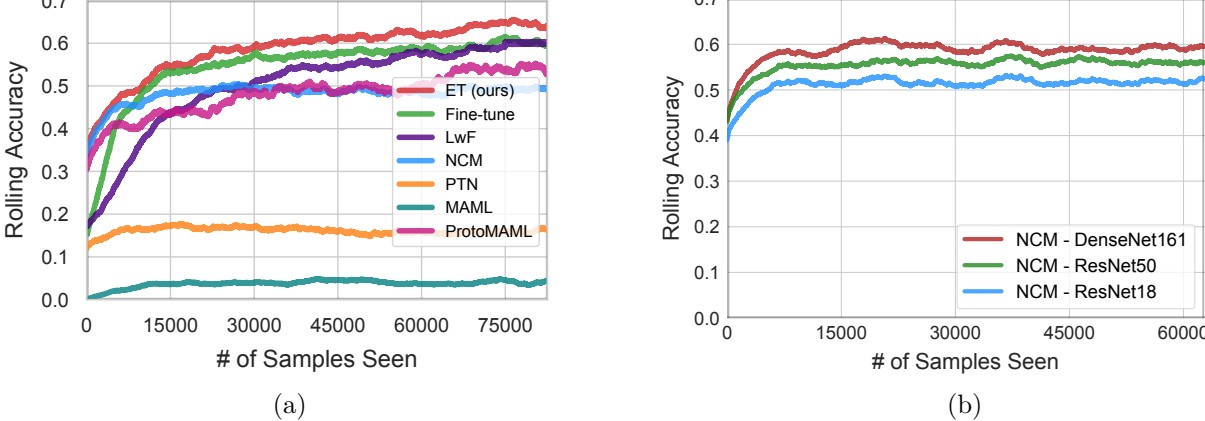

Figure 2: (a) Compares the accuracy of various methods over the stream of data. (b) Compares the accuracy of NCM on novel classes across network architectures. Contrary to prevailing thought, we find that deeper networks generalize better to novel few-shot classes.

few-shot works indicate that training with deeper and overparameterized networks decrease performance (Sun et al., 2019; Oreshkin et al., 2018; Snell et al., 2017; Finn et al., 2017). We verify this observation, noting that the 4-layer convnet PTN (Table 3-a) outperforms the ResNet18 PTN in overall and novel class accuracy.

The prevailing thought in few-shot literature has been that smaller networks overfit less to base classes, and therefore methods use shallow networks or develop techniques to constrain deeper ones. We find evidence to the contrary, that deeper networks generalize better to novel classes when using standard batch sampling (see

Figure 2-b). Given that NCM and PTN differ only in the use of meta-training, the experiments indicate that meta-training is responsible for the lower performance of deeper networks. This evidence is further reinforced by the fact that Meta-Baseline performs worse than NCM with the inclusion of meta-training (Table 3-e,p).

For the more recent few-shot methods (ProtoMAML, Meta-Baseline, ConstellationNet) that utilize a combination of meta-training and mini-batch training we observe significantly better results compared to purely meta-trained methods. However, we find that these recent few-shot methods are outperformed by the NCM baseline for novel, pretrain, tail and head classes (Table 3) indicating that meta-training in its current form reduces performance in settings with varying number of examples.

Typically meta-training is performed with a fixed number of classes (way) and number of examples (shot). In FLUID the number of examples and classes are changing, therefore standard meta-training may not be suitable to match the test conditions. We conduct experiments to observe whether changing the meta-training procedure to better reflect the test conditions improves performance of the model. We ablate over the way and shot number as well as randomly sample the shot and way throughout training. Results can be found in Appendix P. We find that changing the shot hyper-parameter changes which part of the distribution the model performs well on (tail and head), but the overall accuracy does not significantly improve.

## 5.2 Continual Learning Analysis

We evaluate Learning without Forgetting (LwF), Elastic Weight Consolidation (EWC), Dark Experience Replay (DER), and Experience Replay with Asymmetric Cross Entropy (ER-ACE). For analysis, we compare to the baselines NCM, standard training, and fine-tuning. LwF and EWC are prominent CL methods while DER and ER-ACE are recent methods with state-of-the-art performance.

We find that catastrophic forgetting of the pretrain parameters is a significant challenge in the FLUID evaluation. Standard training of the parameters on the sequential data degrades not only the accuracy for pretrain classes, but also for novel classes compared to CL methods and freezing network parameters (table 3). We hypothesize that large-scale pretraining provides better features even for novel classes compared to training on the smaller sequential data set. A similar observation was made by Hayes & Kanan (2020) in a CL setting. Further evidence for this can be found in the update strategies section where we observe that standard training on sequential data for too many epochs reduces overall accuracy (Section 5.6 - Figure 3).

Our experiments show that freezing the feature extractor (NCM and fine-tuning) is more effective in preventing catastrophic forgetting than existing CL methods. Specifically, NCM obtains $\sim 8\%$ (Table 3 j-m) higher mean-per-class accuracy compared to existing CL methods and fine-tuning obtains $\sim 4\%$ higher overall accuracy. This result indicates that there is significant room for progress in reducing forgetting in the FLUID setting and motivates the need for new evaluations which more closely model the challenges faced by real-world ML systems. For scenarios in which the pretraining data is radically different from the target distribution the above conclusion may not hold, such as for permuted MNIST.

The notable differences between FLUID and standard CL formulations are the inclusion of pretraining, flexible training, one distribution shift rather than multiple, and measuring performance only on the deployment distribution. We contend pretraining is a reasonable inclusion as real-world vision systems have access to large datasets such as ImageNet, though some scenarios, especially those outside the domain of computer vision, may not afford pretraining.

## 5.3 Exemplar Tuning

We find that ET (Table 3-w) has significantly higher overall and mean-class accuracy than other evaluated methods and uses similar compute as fine-tuning. Figure 2-a shows how ET quickly adapts to new classes and continues to learn in the standard data regime (high accuracy at the start and end of the stream). Finally, we show that ET outperforms simple NCM + fine-tuning (Weight Imprinting) by $\sim 10\%$, in addition to the practical advantages outlined in section 4.

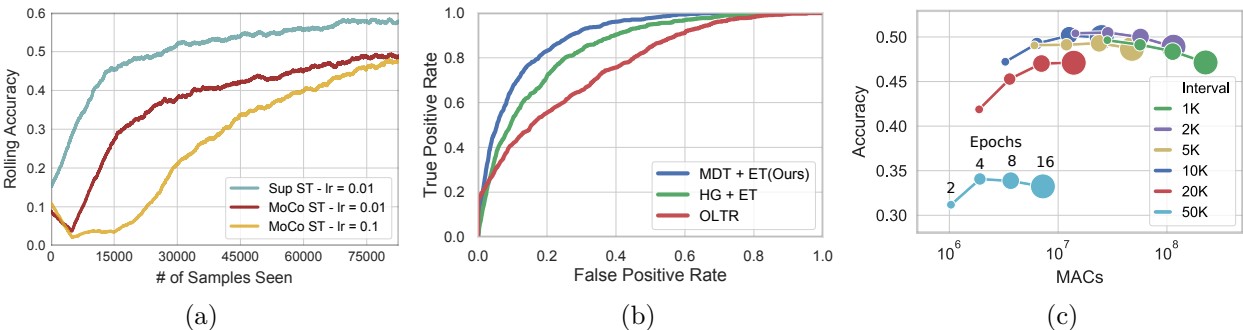

Figure 3: (a) Accuracy of standard training with MoCo & supervised pretraining. Unexpectedly, MoCo accuracy falls during the initial streaming phase. (b) ROC curves for unseen class detection. MDT outperforms all OOD baselines evaluated in FLUID. (c) Standard training accuracy curve for a range of training frequencies & epochs showing that over training can lead to lower accuracy. MACs $\propto$ total gradient updates.

## 5.4 Novel Class Detection and MDT

We measure AUROC for detecting new classes throughout the sequence and present in Figure 3-b. HG baseline + ET, OLTR, and MDT + ET achieve 0.84, 0.78 and 0.92 AUROC scores respectively. The performance of Minimum Distance Thresholding (MDT) indicates that standard recognition networks are well suited for detecting out-of-distribution classes and can be done simultaneously with classification. We compare MDT and HG baseline with other classifiers such as NCM and fine-tuning in Appendix L.

## 5.5 Representation Learning Analysis

We observe unexpected behavior from contrastive MoCo (He et al., 2020b) and VINCE (Gordon et al., 2020) representations in the FLUID setting. For fine-tuning the classifier of a MoCo representation, we find that accuracy is less than 1% and 4.96% on novel and pretrain tail classes respectively (Table 3-t). In comparison the supervised counterpart (Table 3-n) obtains 23.56% and 58.77% accuracy respectively. We conclude that this difficulty is due to learning the linear classifier because NCM with MoCo (Table 3-v) does not exhibit the same drop in performance. Figure 3-a shows other unexpected behavior in which MoCo accuracy drastically decreases initially when standard training, then begins improving after 10K samples. This behavior is not observed for supervised pretraining and occurs for a range of learning rates. We argue that this is related to learning a mixture of pretrain and novel classes which is the primary difference between FLUID and previous downstream tasks. The significantly lower accuracy of MoCo representations on novel-tail classes while fine-tuning (Table 3-t) further reinforces this hypothesis. These observations and insights are also observed for VINCE (Gordon et al., 2020), a similar contrastive method (see Appendix G). These results validate the utility of an evaluation such as FLUID which assess the capabilities of methods more generally.

## 5.6 Update Strategies

We investigate trade-offs between compute cost and accuracy of simple update strategies and leave the challenges of learning adaptive update strategies in FLUID as future work.

For update strategies, we ablate over the frequencies and number of training epochs per update (Figure 3-c & 6) and measure the trade-off in accuracy and total compute cost. We conduct these experiments for fine-tuning (Figure 6 in Appendix O) and for standard training (Figure 3-c) on ResNet18 model with supervised pretraining.

We observe that training for too many total epochs (training frequency × epochs) with standard training (Figure 3-c) decreases overall accuracy, though for fine-tuning accuracy asymptotically improves (Figure 6 in Appendix O). We hypothesize that the optimal amount of training balances the features learned from ImageNet-1K with those from the smaller, imbalanced streaming data. This aligns with our continual learning

experiments that indicate large-scale pretrained features trained on more data outperform specialized features. These initial experiments are intended to illustrate the new problems that FLUID presents for future research. The results indicate that there is significant room for improvement in both efficiency and accuracy with new strategies for training networks under streaming conditions which we leave for future work.

### 5.7 Summary of Insights

1. Representative few-shot methods do not scale well to FLUID with a varying number of examples per class and more classes. **Supporting Evidence:** Prototypical Networks (PTN) and MAML perform 30% worse in overall accuracy compared to baselines of fine-tuning and nearest class mean (NCM) (Table 3 a-b, n-p). Recent state-of-the-art methods ProtoMAML, SimpleCNAPS, and ConstellationNet are outperformed by the baseline NCM for both novel and pretrain classes.

2. The current formulation of meta-training inhibits few-shot methods from scaling to more data and larger architectures in the FLUID setting. **Supporting Evidence:** PTN decreases in all accuracy metrics when increasing architecture size from Conv-4 to ResNet18 (Table 3 a, c) while NCM increases in all accuracy metrics with larger models (Fig 2b). PTN & NCM differ only in that PTN uses meta-training while NCM uses standard batch training.

3. Catastrophic forgetting is a significant challenge in the FLUID setting which is not solved by existing continual learning approaches and large-scale pretraining changes the types of methods which are effective for preventing forgetting. **Supporting Evidence:** Freezing the network parameters (Fine-tuning and NCM) obtains higher accuracy on novel and pretrain classes compared to all evaluated CL methods (Table 3).

4. Contrastive self-supervised representations perform significantly worse on novel classes compared to those of supervised when learning from a mix of novel and pretrain classes in FLUID. **Supporting Evidence:** Fine-tuning from MoCo (Table 3) and VINCE (Appendix Table 6) obtain 0.1% and 1.69% accuracy on novel tail classes. Supervised fine-tuning obtains 23.56% on the same cross-section of data. This drastic gap between supervised & MoCo representation is absent in the original work by He et al. (2020b) when fine-tuning to COCO and other downstream tasks.

## 6 Limitations and Future Work

Throughout this paper, we studied various methods and settings in the context of supervised image classification, a highly explored problem in ML. While we do not make design decisions specific to image classification, incorporating other mainstream tasks into FLUID is a potential next step. Also while the FLUID framework is agnostic to any particular data set, our experiments and conclusions are anchored in the computer vision domain. Across the experiments in this paper, we impose some assumptions about the learning conditions, albeit only a few, on FLUID. For example, we currently assume that FLUID has access to labels as the data streams in. One exciting future direction is to add the semi- or un-supervised aspects to FLUID. Relaxing these remaining assumptions to bring FLUID closer to real world conditions is an interesting future direction.

## 7 Conclusions

We introduce FLUID, a unified evaluation framework designed to facilitate research towards more general methods capable of handling the challenges of learning in deployment settings. FLUID enables comparison and integration of solutions across few-shot, transfer, continual and representation learning, & out-of-distribution detection while introducing new research challenges like how and when to update model parameters based on incoming data. Through our experiments with FLUID on a wide-range of methods we show the limitations and merits of existing solutions. For example, few-shot methods do not scale well to settings with more classes and varying number of examples and freezing network parameters prevents catastrophic forgetting better than representative continual learning methods when in the FLUID setting. As a starting point for solving the new challenges, we present two baselines, Exemplar Tuning & Minimum Distance Thresholding, which outperform existing methods on FLUID.

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

## A  FLUID **Procedure**

Algorithm 1 describes the high level implementation directives of FLUID framework.

---

**Algorithm 1** FLUID Procedure

---

**Input:** Task $\mathcal{T}$
**Input:** ML sys.: (pretrained) model $\mathbf{f}$, update strategy $\mathbf{S}$
**Output:** Evaluations: $E$, Operation Counter: $C$
 1: **function** FLUID($\mathcal{T}, (\mathbf{f}, \mathbf{S})$)
 2:     Evaluations $E = [\,]$
 3:     Datapoints $\mathcal{D} = [\,]$
 4:     Operation Counter $C = 0$.

 5:     **while** streaming **do**
 6:         Sample $\{x, y\}$ from $\mathcal{T}$
 7:         prediction $p = \mathbf{f}(\mathbf{x})$ ($A$ operations)
 8:         Flag $n$ indicates if $y$ is a new unseen class
 9:         $E$.insert($\{y, p, n\}$)
10:         $\mathcal{D}$.insert($\{x, y\}$)
11:         Update $\mathbf{f}$ using $\mathbf{S}$ with $\mathcal{D}$ ($B$ operations)
12:         $C += A + B$
13:     **end while**

14:     **return** $E, C$
15: **end function**

---

## B  Dataset Information

The five sequences we pair with FLUID are constructed from ImageNet-22K (Deng et al., 2009). Two sequences (1-2) are for validation, and three (3-5) are for testing. Each sequence contains 1,000 classes; 250 of which are in ImageNet-1K (Russakovsky et al., 2015) (pretrain classes) and 750 of which are only in ImageNet-22K (novel classes). For the test sequences, we randomly select the classes without replacement to ensure that the sequences do not overlap. The validation sequences share pretrain classes because there are not enough pretrain classes (1000) to partition among five sequences. We randomly distribute the number of images per class according to Zipf's law with $s = 1$ (Figure 4). For classes without enough images, we fit the Zipfian distribution as closely as possible which causes a slight variation in sequence statistics seen in Table 4.

Table 4: Statistics for the sequences of images used in FLUID. Sequences 1-2 are for validation and Sequence 3-5 are for testing. The images from ImageNet-22k are approximately fit to a Zipfian distribution with 250 classes overlapping with ImageNet-1k and 750 new classes.

| Sequence # | Number of Images | Min # of Class Images | Max # of Class Images |
|:---:|:---:|:---:|:---:|
| 1 | 89030 | 1 | 961 |
| 2 | 87549 | 21 | 961 |
| 3 | 90133 | 14 | 961 |
| 4 | 86988 | 6 | 892 |
| 5 | 89921 | 10 | 961 |

## C  Dataset License

ImageNet does not explicitly provide a license.

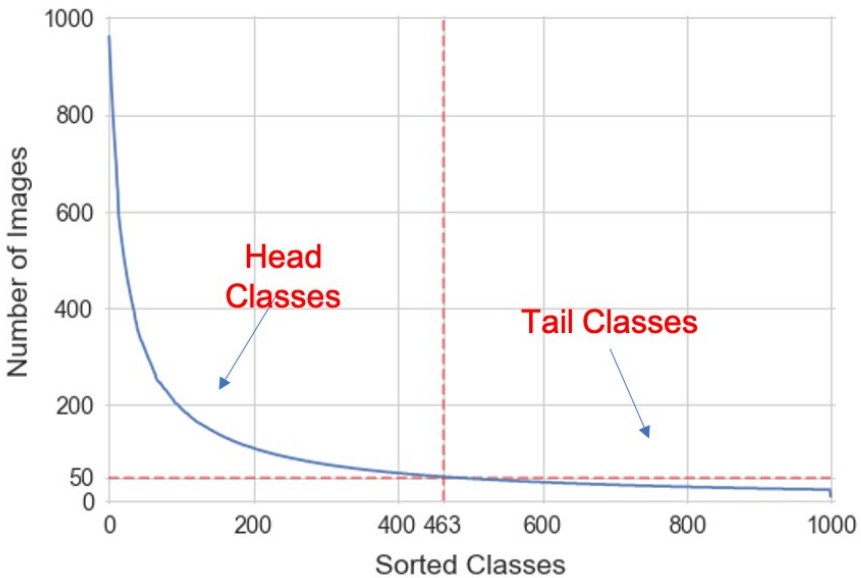

Figure 4: The distribution of samples over the classes for Sequences 1 - 5. Classes with less than 50 samples are considered in the tail and samples with greater than or equal to 50 samples are considered in the head for the purpose of reporting.

## D    More Method Details

**Nearest Class Mean**   Each class mean, $m_i$, is the average feature embedding of all examples in class $i$: $m_i = \sum_{x \in C_i} f_\phi(x)$; where $C_i$ is the set of examples belong to class $i$ and $f_\phi$ is the deep feature embedding of $x$. Class probabilities are the softmax of negative distances between $x$ and class means:

$$P(y = i | \mathbf{x}) = \frac{e^{-d(m_i, f_\phi(\mathbf{x}))}}{\sum_{i'} e^{-d(m_{i'}, f_\phi(\mathbf{x}))}} \tag{2}$$

**MAML**   The gradient update for MAML is: $\theta \leftarrow \theta - \beta \cdot \nabla_\theta \sum_{\mathcal{T}_i \sim p(T)} \mathcal{L}_{\mathcal{T}_i} \left( f_{\theta_i'} \right)$ where $\theta_i'$ are the parameters after making a gradient update given by: $\theta_i' \leftarrow \theta - \alpha \cdot \nabla_\theta \mathcal{L}_{\mathcal{T}_i} (f_\theta)$.

**OLTR**   The network consist of two parts 1) A feature extractor consist of a ResNet backbone followed by a modulated attention and 2) A classifier and memory bank that are used to classify the output of the feature extractor. Training is done in 2 stages; In the first stage the feature extractor is trained. In the second stage the feature extractor and classifier are fine-tuned while samples are accumulated in the memory bank.

**Weight Imprinting**   Weight Imprinting initializes the weights of the cosine classification layer, then performs fine-tuning using all of the data with a learnable temperature to scale the logits. Weight Imprinting can be thought of as NCM with cosine similarity as the metric for determining the closest neighbor, then performing fine-tuning. To use Weight Imprinting in a sequential setting, rather than a few-shot setting, we must decide when to begin fine-tuning. In the original formulation, fine-tuning was performed after the centroids were calculated using the entire data set, but in the sequential setting we do not have access to the entire data set until streaming ends. Therefore we choose to begin fine-tuning when the accuracy of fine-tuning exceeds NCM on validation data. In a real-world scenario it would be difficult to obtain such information, but we employ such a strategy to provide an upper-bound for the performance of Weight Imprinting in the sequential setting.

**ProtoMAML**   ProtoMAML initializes the classification layer as nearest class mean in accordance with Prototypical Networks, then performs second order updates according to the MAML objective. Weight Imprinting and ProtoMAML differ in that ProtoMAML is trained with meta-training after it is initialized

with the batch trained backbone. We use the first order variant in accordance with (Triantafillou et al., 2020) for computational efficiency.

## E    Implementation Details

In this section, we discuss how methods are adapted with respect to FLUID. Some methods are intuitively applied with little modification, and some require interpretation for how they should be adapted.

**Offline Training**  For all experiments (Table 3) that require offline training (fine-tuning, Weight Imprinting, standard training, ET and LwF), except OLTR, we train each model for 4 epochs every 5,000 samples observed. An epoch includes training over all previously seen data in the sequence. Experiments in Figure 6 show that training for 4 epochs every 5,000 samples balanced sufficient accuracy and reasonable computational cost. Fine-tuning experiments use a learning rate of 0.1 and standard training uses 0.01 for supervised pretraining. For MoCo pretraining fine-tuning uses a learning rate of 30 and standard training uses 0.01. All the experiments use the SGD+Momentum optimizer with a 0.9 momentum.

**Instance-Based Updates**  All instance-based methods (NCM, ET, Weight Imprinting, Prototypical Networks) are updated after every sample as it takes no additional compute compared to batch updates.

**Meta-Training**  For Prototypical-Networks and MAML we meta-train from scratch with the n-shot k-way paradigm. We use 5-shot 30-way in accordance with the original works (Snell et al., 2017) (Finn et al., 2017). We trained according We meta-train for 100 epochs with a learning rate of 0.01 and reduce it by 0.5 every 40 epochs. For ProtoMAML and SimpleCNAPS we train according to the Meta-Dataset routine (Triantafillou et al., 2020). Meta-Dataset splits the 1000 classes into train, validation, and test. We meta-train with all 1000 classes for fair comparison and sample between 5 and 50 classes in accordance with the original work. ConstellationNet training consists of both standard batch and meta-training. We train with 5-shot 30-way with the hyper-parameters provided in their codebase (Xu et al., 2020).

**Exemplar Tuning**  We initialize the residual vectors as zero. ET is trained according to the specifications of instance-based updates and offline training simultaneously.

**Weight Imprinting**  For Weight Imprinting, we transition from NCM to fine-tuning after 10,000 samples as we observed that the accuracy of NCM saturated at this point in the validation sequence. We use a learning rate of 0.1 while fine-tuning.

**Learning Without Forgetting**  We adapt Learning Without Forgetting to the FLUID task by freezing a copy of the model after pretraining which is used for knowledge distillation. Not all pretraining classes are seen during streaming so only softmax probabilities for classes seen during the stream are used in the cross-entropy between the soft labels and predictions. We use a temperature of 2 to smooth the probabilities in accordance with (Li & Hoiem, 2017). We swept over $\lambda_o$ values between $\{.1, .2, \ldots 1\}$. We found .2 maximized mean per-class and overall accuracy. Training is done according to the specifications given in the offline training portion of this section.

**Elastic Weight Consolidation**  We adapt Elastic Weight Consolidation (Kirkpatrick et al., 2017) to the FLUID task by freezing a copy of the model after pretraining as the optimal model of the pretrain task. We then use the validation set of ImageNet-1K corresponding to the 250 classes being used for the computation of Fisher information per-parameter. For every training step in FLUID, a penalty is added based on the distance moved by the parameters from the base model weighted by the Fisher information. The Fisher information is calculated at the start of every flexible train step to mitigate catastrophic forgetting. Training is done according to the specifications given in the offline training portion of this section. Depending on how frequently the Fisher information is computed, the compute associated increases over the standard training costs. We use the default hyper-parameters of .1 for $\lambda$ and .9 for $\alpha$ where the loss is: $\mathcal{L}(\theta) = \alpha \mathcal{L}_B(\theta) + \sum_i \frac{\lambda}{2} F_i \left(\theta_i - \theta^*_{A,i}\right)^2$.

**DER and ER-ACE**  We train DER and ER-ACE with the standard specifications reported in the offline training portion of section E. For DER, at each offline training phase we store the logits and use them to compute the KL divergence term in the next offline training phase. For $\alpha$ we swept over values between $\{.2, .4, \ldots 1\}$ and found .5 to be most effective.

For ER-ACE we apply the loss given in the paper:

$$\mathcal{L}\_ace \left(\mathbf{X}^{bf} \cup \mathbf{X}^{in}\right) = \mathcal{L}\_ce(\mathbf{X}^{bf}, C\_old \cup C\_curr\ ) + \mathcal{L}\_ce(\mathbf{X}^{in}, C\_curr)$$

Where $C_{old}$ contains the 1000 pretraining classes and $C_{curr}$ contains the novel classes which have been seen during streaming.

Both methods are performed at the end of each offline training routine. The replay buffer size for both methods is 90,000 for fair comparison to other methods. In Table 14 we compare the baseline NCM with DER and ER-ACE with smaller replay buffer sizes. Note the NCM baseline uses no replay buffer while other CL methods do use a replay buffer to store data during the sequential phase.

**OLTR** For OLTR (Liu et al., 2019), we update the memory and train the model for 4 epochs every 200 samples for the first 10,000 samples, then train 4 epochs every 5,000 samples with a 0.1 learning rate for classifier parameters and 0.01 for feature extraction parameters which is in accordance with the specifications of the original work.

**Pretraining** We use the PyTorch (Paszke et al., 2019) ResNet18 and ResNet50 models pretrained on supervised ImageNet-1K. We use the models from Gordon et al. (Gordon et al., 2020) for the MoCo (He et al., 2020b) self-supervised ImageNet-1K pretrained models. MoCo-ResNet18 and MoCo-ResNet50 get top-1 validation accuracy of 44.7% and 65.2% respectively and were trained for 200 epochs. For fine-tuning and ET with MoCo, we report the results with a learning rate of 30 which is suggested by the original work when learning on frozen features. All other learning rates with MoCo are the same as with supervised.

## F  Training Depth for Fine Tuning

We explored how training depth affects the accuracy of a model on new, old, common, and rare classes. For this set of experiments, we vary the number of trained layers when fine-tuning for 4 epochs every 5,000 samples on ResNet18 with a learning rate of 0.01 on Sequence 2 (validation). The results are reported in Table 5. We found that training more layers leads to greater accuracy on new classes and lower accuracy on pretrain classes. However, we observed that the number of fine-tuning layers did not significantly affect overall accuracy so for our results on the test sequences (3-5) we only report fine-tuning of one layer (Table 3).

Table 5: The results for fine-tuning various numbers of layers with a learning rate of .01 on Sequence 2. Training more layers generally results in higher accuracy on novel classes, but lower accuracy on pretrain classes. The trade-off between novel and pretrain accuracy balances out so the overall accuracy is largely unaffected by the depth of training.

| # of Layers | Novel-Head (>50) | Pretrain-Head (>50) | Novel-Tail (<50) | Pretrain-Tail (<50) | Mean Per-Class | Overall |
|---|---|---|---|---|---|---|
| 1 | **41.32** | **80.96** | 17.13 | 66.52 | 39.19 | 56.87 |
| 2 | 41.55 | 80.79 | 17.40 | **67.03** | 39.43 | 56.79 |
| 3 | 45.82 | 78.59 | 19.08 | 59.52 | **40.73** | **57.23** |
| 4 | **46.96** | 75.44 | 19.87 | 53.97 | 40.39 | 57.04 |
| 5 | 46.76 | 75.72 | **19.97** | 54.04 | 40.41 | 57.04 |

## G  Results for VINCE ResNet18 backbone on Sequence 5

We report all performance metrics for sequence 5 in Table 6 for ResNet18 backbone trained via VINCE (Gordon et al., 2020). VINCE is a self-supervised representation learning method that focuses on leveraging video as a natural form of augmentation for contrastive learning. These results corroborate the findings of Table 3 which uses ResNet18 backbone trained via MoCo (He et al., 2020b) further solidifying the insights drawn on self-supervised representation learning methods.

Table 6: Results on sequence 5 with ResNet18 backbone trained using VINCE (Gordon et al., 2020)

| Method | Pretrain Strategy | Novel - Head (>50) | Pretrain - Head (>50) | Novel - Tail (<50) | Pretrain - Tail (<50) | **Mean Per-Class** | **Overall** | GMACs↓ ($\times 10^6$) |
|---|---|---|---|---|---|---|---|---|
| | | | Backbone - ResNet18 | | | | | |
| (a) Fine-tune | VINCE | 18.00 | 14.61 | 1.89 | 1.56 | 7.25 | 26.27 | 0.16 / 5.73 |
| (b) Standard Training | VINCE | 24.60 | 32.06 | 6.38 | 9.63 | 16.17 | 32.95 | 11.29 |
| (c) NCM | VINCE | 15.96 | 22.32 | **12.32** | **16.08** | 15.20 | 18.28 | **0.15** |
| (d) **Exemplar Tuning** | VINCE | **26.84** | **37.03** | 7.32 | 11.30 | **18.11** | **35.44** | 0.16 / 5.73 |

## H   Results for ResNet50 backbone on Sequence 5

We report all performance metrics for sequence 5 in Table 7 for ResNet50 backbone. These results corroborate the findings of Table 3 which uses ResNet18 backbone.

Table 7: Continuation of Table 3 results on sequence 5 with ResNet50 backbone.

| Method | Pretrain Strategy | Novel - Head (>50) | Pretrain - Head (>50) | Novel - Tail (<50) | Pretrain - Tail (<50) | **Mean Per-Class** | **Overall** | GMACs↓ ($\times 10^6$) |
|---|---|---|---|---|---|---|---|---|
| | | | Backbone - ResNet50 | | | | | |
| (a) Fine-tune | MoCo | 14.42 | 43.61 | 0.22 | 13.40 | 11.85 | 31.35 | 0.36 / 13.03 |
| (b) Fine-tune | Sup. | 47.78 | 82.06 | 27.53 | **66.42** | 46.24 | 57.95 | 0.36 / 13.03 |
| (c) Standard Training | MoCo | 26.82 | 42.12 | 10.50 | 21.08 | 21.32 | 35.44 | 38.36 |
| (d) Standard Training | Sup. | 43.89 | 74.50 | 21.54 | 50.69 | 39.48 | 54.10 | 38.36 |
| (e) NCM | MoCo | 30.58 | 55.01 | 24.10 | 45.37 | 32.75 | 36.14 | **0.35** |
| (f) NCM | Sup. | 45.58 | 78.01 | **35.94** | 62.90 | 47.75 | 52.19 | **0.35** |
| (g) LwF | Sup. | 21.52 | 49.17 | 5.49 | 38.74 | 20.69 | 30.57 | 38.36/76.72 |
| (h) EWC | Sup. | 43.84 | 76.03 | 21.22 | 53.64 | 39.89 | 54.59 | 40.36 |
| (i) **Exemplar Tuning** | MoCo | 28.86 | 54.03 | 7.02 | 20.82 | 21.89 | 40.13 | 0.36 / 13.03 |
| (j) **Exemplar Tuning** | Sup. | **52.95** | **82.27** | 28.13 | 57.15 | **48.02** | **62.41** | 0.36 / 13.03 |

## I   Results For Other Sequences

We report the mean and standard deviation for all performance metrics across test sequences 3-5 in Table 8. Note that the standard deviation is relatively low so the methods are consistent across the randomized sequences.

## J   Results for FLUID - Places365

We reproduced the experiments on a long-tailed version of Places365 and find results consistent with those on the FLUID variant of ImageNet. Results can be found in table 9

## K   Prototypical Network Experiments

We benchmarked our implementation of Prototypical Networks on few-shot baselines to verify that it is correct. We ran experiments for training on both MiniImageNet (Vinyals et al., 2016) and regular ImageNet-1k and tested our implementation on the MiniImageNet test set and FLUID (Sequence 2). We found comparable results to those reported by the original Prototypical Networks paper (Snell et al., 2017) (Table 10).

## L   Out-of-Distribution Ablation

In this section we report AUROC and F1 for MDT and softmax for all baselines. In section 5 we only included OLTR, MDT with Exemplar Tuning, and ET with maximum softmax (Hendrycks Baseline). Additionally,

Table 8: Averaged results for all methods evaluated on Sequences 3-5. See Table 3 for the computational cost (GMACs) for each method and more information about each column.

| Method | Pretrain | Novel - Head (>50) | Pretrain - Head (>50) | Novel - Tail (<50) | Pretrain - Tail (>50) | **Mean Per-Class** | **Overall** |
|---|---|---|---|---|---|---|---|
| | | Backbone - Conv-4 | | | | | |
| Prototype Networks | Meta | 5.02±0.05 | 9.71±0.11 | 0.64±0.01 | 1.27±0.04 | 3.25±0.03 | 7.82±0.09 |
| MAML | Meta | 2.93±0.01 | 2.02±0.02 | 0.15±0.01 | 0.1±0.01 | 1.11±0.02 | 3.64±0.06 |
| | | Backbone - ResNet18 | | | | | |
| Prototype Networks | Meta | 8.72±0.09 | 16.84±0.14 | 7.06±0.03 | 12.98±0.04 | 9.46±0.08 | 11.19±0.12 |
| Meta-Baseline | Sup./Meta | 41.73±0.57 | 66.54±2.37 | 27.54±1.13 | 53.69±0.97 | 39.32±0.71 | 47.74±0.63 |
| Fine-tune | Moco | 5.31±0.24 | 45.95±1.27 | 0.03±0 | 26.23±0.88 | 10.64±0.23 | 18.52±0.98 |
| Fine-tune | Sup. | 43.2±0.65 | 74.55±2.53 | 22.79±1.21 | 59.63±1.02 | 40.9±0.73 | 53.06±0.65 |
| Standard Training | Moco | 26.9±0.27 | 42.39±3.04 | 9.1±0.74 | 21.11±0.51 | 20.76±0.32 | 34.85±0.75 |
| Standard Training | Sup. | 38.82±0.49 | 65.88±2.32 | 16.15±0.83 | 44.3±0.91 | 33.63±0.38 | 48.81±0.57 |
| NCM | Moco | 19.31±0.06 | 30.02±1.69 | 14.21±0.46 | 22.06±0.52 | 18.86±0.13 | 22.14±1.24 |
| NCM | Sup. | 41.68±0.65 | 70.05±2.29 | 31.24±0.86 | 57.23±0.97 | 42.87±0.62 | 47.89±0.76 |
| OLTR | MoCo | 41.47±0.03 | 31.48±0.01 | 17.48±0.01 | 9.81±0.01 | 22.03±0 | 38.33±0.01 |
| OLTR | Sup. | 51.19±0.37 | 37.02±0.51 | 24.14±0.14 | 13.77±0.24 | 27.6±0.28 | 44.46±0.44 |
| **Exemplar Tuning** | Moco | 32.57±1.54 | 43.48±0.4 | 6.39±0.49 | 12.81±0.12 | 18.46±0.35 | 39.25±1.20 |
| **Exemplar Tuning** | Sup. | 46.36±2.31 | 69.34±0.53 | 23.48±1.23 | 45.82±0.32 | 42.93±0.17 | 57.56±0.56 |
| | | Backbone - ResNet50 | | | | | |
| Fine-tune | Moco | 45.95±0.26 | 5.31±0.32 | 26.23±0.07 | 0.03±1.74 | 10.64±0.21 | 18.52±1.02 |
| Fine-tune | Sup. | 47.59±0.65 | 80.14±1.71 | 26.69±0.97 | 66.92±1.4 | 45.62±0.6 | 57.48±0.47 |
| Standard Training | Moco | 43.93±0.73 | 71.72±3.18 | 20.84±0.92 | 51.43±0.68 | 38.94±0.9 | 53.45±1.73 |
| Standard Training | Sup. | 47.59±0.45 | 80.14±2.59 | 26.69±0.79 | 66.92±1.91 | 45.62±0.47 | 57.48±0.56 |
| NCM | Moco | 30.15±0.48 | 53.84±1.05 | 23.99±0.53 | 44.11±1.11 | 32.27±0.92 | 35.45±0.61 |
| NCM | Sup. | 45.46±0.95 | 76.55±1.77 | 35.47±0.82 | 65.62±1.57 | 47.77±0.65 | 52.22±0.55 |
| **Exemplar Tuning** | Moco | 28.46±3.04 | 40.42±1.33 | 7.57±2.15 | 14.36±4.14 | 19.54±2.63 | 32.07±2.37 |
| **Exemplar Tuning** | Sup. | 49.24±1.55 | 75.78±1.84 | 26.67±2.17 | 55.63±2.31 | 44.15±1.44 | 62.35±1.02 |

Table 9: The FLUID evaluation applied to Places365 with ResNet18 architecture. ET outperforms baselines in overall and mean-per-class (MPC) acc.

| Method | Novel Head | Pretrain Head | Novel Tail | Pretrain Tail | MPC Acc. | Acc. |
|---|---|---|---|---|---|---|
| NCM | 24.2 | 71.1 | **16.9** | 71.1 | 24.1 | 41.4 |
| PM [8] | 25.1 | 81.3 | 8.9 | 82.3 | 25.2 | 49.8 |
| Fine-Tune | 25.1 | 83.8 | 7.8 | 83.8 | 24.7 | 50.0 |
| ET (Ours) | **29.9** | **84.6** | 10.5 | **84.6** | **27.8** | **54.6** |

we visualize the accuracy curves for in-distribution and out-of-distribution samples as the rejection threshold vary (Figure 5). All the OOD experiments presented in Figure 5 and Table 11 were run using ResNet18. Minimum Distance Thresholding (MDT) threshold distances but also similarity metrics can be used. MDT generally works better than maximum softmax when applied to most methods.

The results of NCM and Exemplar Tuning using softmax and dot product similarity in comparison to OLTR are shown in table 11. The F1-scores are low due to the large imbalance between positive and negative classes. There are 750 unseen class datapoints vs $\sim 90000$ negative datapoints. Table 11 shows that cosine similarity (MDT) is better than softmax or the OLTR model for most methods.

Table 10: Our implementation of Prototypical Networks on MiniImageNet & Fluid. $^\diamond$ Results from Snell et al. (2017).

| Method | Backbone | Train Set | MiniImageNet 5 Way - 5 Shot | Fluid |
|---|---|---|---|---|
| Prototypical Networks | Conv - 4 | MiniImageNet | 69.2 | 14.36 |
| Prototypical Networks | Conv - 4 | ImageNet (Train) | 42.7 | 15.98 |
| Prototypical Networks$^\diamond$ | Conv - 4 | MiniImageNet | 68.2 | - |

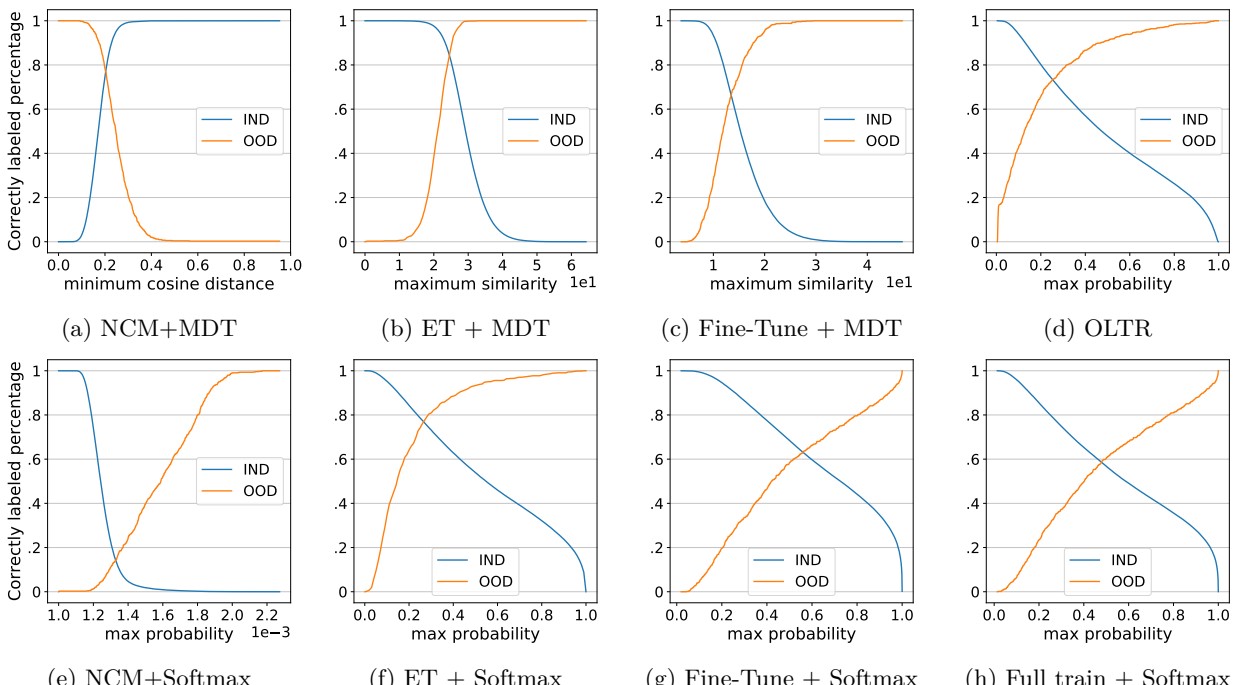

(a) NCM+MDT     (b) ET + MDT     (c) Fine-Tune + MDT     (d) OLTR

(e) NCM+Softmax     (f) ET + Softmax     (g) Fine-Tune + Softmax     (h) Full train + Softmax

Figure 5: The accuracy for the in-distribution (IND) and out-of-distribution (OOD) samples as the threshold for considering a sample out-of-distribution varies. The horizontal axis is the threshold value, and the vertical axis is the accuracy. Intersection of the IND and OOD curves at a higher accuracy generally indicates better out-of-distribution detection for a given method.

Table 11: The out-of-distribution performance for each method on sequence 5. We report the AUROC and the F1 score achieved by choosing the best possible threshold value.

| Metric | NCM +Softmax | NCM +MDT | Exemplar Tuning +Softmax | Exemplar Tuning +MDT | Standard Training +Softmax | Standard Training +MDT | Fine-Tune +Softmax | Fine-Tune +MDT | OLTR |
|---|---|---|---|---|---|---|---|---|---|
| AUROC | 0.07 | 0.85 | 0.84 | 0.92 | 0.59 | 0.53 | 0.68 | 0.72 | 0.78 |
| F1 | 0.01 | 0.20 | 0.10 | 0.20 | 0.03 | 0.02 | 0.06 | 0.10 | 0.27 |

# M   Weight Imprinting and Exemplar Tuning Ablations

In Table 13, we ablate over various softmax temperature initializations with Weight Imprinting. We learn the temperature as described in (Qi et al., 2018), but find that initial value affects performance. We report the best results in the main paper. We also ablate over the similarity metrics use in ET. We find that the dot product (linear) is the best measure of similarity for ET.

Table 12: Comparison of ER-ACE and DER with NCM for varying buffer sizes. ER-ACE and DER are designed for smaller buffer sizes therefore we compare these methods against the NCM baseline which uses a buffer size of 0.

| Method | Pretrain | Backbone | Novel - Head (>50) | Pretrain - Head (>50) | Novel - Tail (<50) | Pretrain - Tail (>50) | Mean Per-Class | Overall |
|---|---|---|---|---|---|---|---|---|
| DER (buffer = 5000) | Sup | R18 | 33.93 | 71.57 | 8.75 | 49.55 | 26.85 | 46.35 |
| AR-ACE (buffer = 5000) | Sup | R18 | 31.49 | 69.83 | 13.52 | 48.11 | 29.20 | 41.71 |
| DER (Buffer = 10000) | Sup | R18 | 34.51 | 72.79 | 9.61 | 50.94 | 30.39 | 47.79 |
| ER-ACE (Buffer = 10000) | Sup | R18 | 32.15 | 70.52 | 14.26 | 48.94 | 32.23 | 44.38 |
| NCM (Buffer = 0) | Sup | R18 | 42.35 | 75.70 | 31.72 | 56.17 | 43.44 | 50.62 |

Table 13: Comparison of Weight Imprinting and Exemplar Tuning with different classifiers and initial temperatures. Exemplar Tuning with a linear layer performs significantly better than all other variants.

| Method | Pretrain | Backbone | Novel - Head (>50) | Pretrain - Head (>50) | Novel - Tail (<50) | Pretrain - Tail (>50) | Mean Per-Class | Overall |
|---|---|---|---|---|---|---|---|---|
| Weight Imprinting (s = 1) | Sup | R18 | 36.58 | 63.39 | 9.32 | 21.80 | 26.85 | 46.35 |
| Weight Imprinting (s = 2) | Sup | R18 | 36.58 | 63.39 | 9.32 | 21.80 | 26.85 | 46.35 |
| Weight Imprinting (s = 4) | Sup | R18 | 40.32 | 67.46 | 15.35 | 34.18 | 32.69 | 48.51 |
| Weight Imprinting (s = 8) | Sup | R18 | 31.18 | 32.66 | 34.77 | 28.94 | 32.56 | 46.67 |
| Exemplar Tuning (Cosine) | Sup | R18 | 33.90 | 18.22 | 4.84 | 1.88 | 11.72 | 31.81 |
| Exemplar Tuning (Euclidean) | Sup | R18 | 43.40 | 66.32 | 21.66 | 42.06 | 37.19 | 51.62 |
| Exemplar Tuning (Linear) | Sup | R18 | **48.85** | **75.70** | **23.93** | **45.73** | **43.61** | **58.16** |

## N  Exemplar Tuning **on Standard Recognition Tasks**

On Mini-ImageNet (Vinyals et al., 2016), for 5-shot 5-way Exemplar Tuning with a ResNet10 backbone obtains an accuracy 72.1% compared to 68.2% for Prototypical Networks. Exemplar Tuning accuracy on ImageNet-LT (Liu et al., 2019) with a ResNet18 backbone is 42.1% while a standard linear layer gets to 41.9%.

## O  Update Strategies

Figure 6 has the accuracy vs MACs trade-off for fine-tuning across various update strategies.

## P  Meta-Training Ablations

Table 14: Ablation over the shot and way number for meta-training during the pretraining stage of FLUID. Randomly sampling indicates uniform sampling over the shot and way between the indicated interval. Models are trained from scratch for 100 epochs on ImageNet-1k with an initial learning rate of .1 and cosine annealing. Random sampling is performed with Prototypical Networks.

| Method | Pretrain | Backbone | Novel - Head (>50) | Pretrain - Head (>50) | Novel - Tail (<50) | Pretrain - Tail (>50) | Mean Per-Class | Overall |
|---|---|---|---|---|---|---|---|---|
| Protopypical Networks (5-shot 20-way) | Sup | R18 | 8.64 | 16.98 | 6.79 | 12.74 | 9.50 | 11.14 |
| Protopypical Networks (5-shot 100-way) | Sup | R18 | 9.23 | 16.71 | 7.67 | 11.48 | 9.46 | 11.05 |
| Protopypical Networks (5-shot 500-way) | Sup | R18 | 8.16 | 15.63 | 7.24 | 11.29 | 9.41 | 10.37 |
| Protopypical Networks (5-shot 1000-way) | Sup | R18 | 7.94 | 14.85 | 5.11 | 9.62 | 7.70 | 8.95 |
| Protopypical Networks (20-shot 100-way) | Sup | R18 | 9.57 | 17.13 | 5.91 | 11.31 | 9.37 | 10.95 |
| Protopypical Networks (50-shot 100-way) | Sup | R18 | 9.82 | 17.26 | 5.17 | 11.02 | 9.18 | 10.71 |
| Protopypical Networks (100-shot 100-way) | Sup | R18 | 9.77 | 16.97 | 5.08 | 10.55 | 8.89 | 10.24 |
| Protopypical Networks (200-shot 100-way) | Sup | R18 | 8.38 | 15.29 | 4.83 | 10.48 | 8.76 | 10.05 |
| Random Sampling (Shot - [1, 100], Way - [20, 100]) | Sup | R18 | 9.19 | 16.85 | 7.38 | 11.88 | 9.44 | 11.19 |
| Random Sampling (Shot - [1, 1000], Way - [2, 1000]) | Sup | R18 | 4.12 | 10.34 | 2.02 | 6.75 | 4.74 | 5.91 |

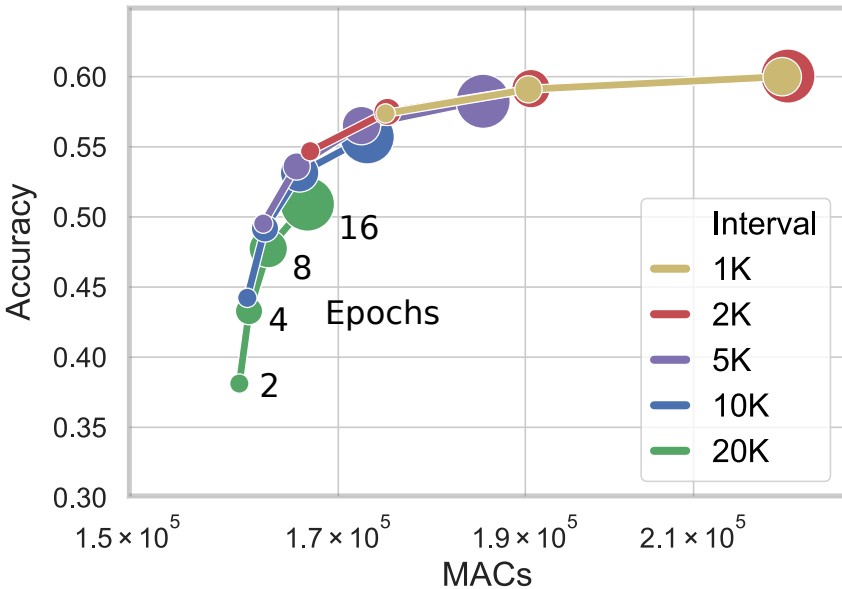

Figure 6: The plot compares the accuracy and MACs for various update strategies when fine-tuning.

## Q   Non-Stationary Distribution

In this section we run preliminary experiments to observe how a non-stationary distribution affects the empirical conclusions about continual learning methods. We alternate every 10,000 samples between the 3 testing sequences which have disjoint novel classes as discussed in section 3 for 100,000 total samples. For these experiments we evaluate on the standard continual learning methods and baselines. We observe that the gap between continual learning methods and baselines is smaller compared to the results found in table 3. In particular, the continual learning methods better prevent forgetting for the pretrain head and tail classes compared to standard training. Overall, we find similar empirical conclusions that pretraining enables methods which freeze the network parameters to outperform traditional continual learning methods.

Table 15: The performance of continual learning methods and baselines evaluated on a non-stationary distribution. All experimental hyper-parameters such as update interval, learning rate, and training epochs are the same as in the original experiments.

| Method | Pretrain Strategy | Novel - Head ($>50$) | Pretrain - Head ($>50$) | Novel - Tail ($<50$) | Pretrain - Tail ($<50$) | **Mean Per-Class** | **Overall** | GMACs↓ ($\times 10^6$) |
|---|---|---|---|---|---|---|---|---|
| | | | | Backbone - ResNet18 | | | | |
| (i) LwF | Sup. | 29.39 | 69.51 | 6.62 | 56.20 | 27.31 | 43.29 | 22.58 / 45.16 |
| (j) EWC | Sup. | 30.03 | 70.17 | 14.80 | 46.34 | 29.84 | 44.19 | 12.29 |
| (k) DER | Sup. | 31.89 | 72.74 | 13.59 | 52.44 | 31.51 | 45.47 | 11.29 |
| (m) Fine-tune | Sup. | 33.51 | 74.75 | 17.99 | 57.17 | 33.75 | 48.73 | 0.16 / 5.73 |
| (n) Standard Training | Sup. | 32.49 | 62.64 | 15.33 | 41.26 | 28.37 | 43.76 | 11.29 |
| (o) NCM | Sup. | 35.25 | 69.27 | 18.97 | 56.67 | 34.02 | 46.13 | 0.15 |
| (p) Exemplar Tuning | Sup. | 36.85 | 73.70 | 19.93 | 45.73 | 35.61 | 49.16 | 0.16 / 5.73 |

## R   FLUID Properties in Real-World Applications

In this section we detail current, real-world applications of machine learning and how they relate to FLUID.

*Computer vision for monitoring biodiversity* (Beery et al., 2018; Tuia et al., 2022). Researchers use camera traps to identify and track animal populations in order to monitor ecological systems. Beery et al. (2018) discuss the challenges of deployment which are similar to those in the FLUID evaluation. Specifically, they cite difficulty in generalizing to new environments, lack of examples for rare classes, and lengthy, intermittent data collection. We've listed the relevant details of monitoring biodiversity as they pertain to FLUID.

- The model is pretrained on wildlife data which differs from the deployment distribution.

- The camera trap systems are deployed and sequentially collect data from the new environment.

- As with many real-world datasets, species observation data exhibits a long-tail distribution (Beery et al., 2018; Sudderth & Jordan, 2008). Rare species must be classified using only a few examples.

- The system may be queried and updated at any time using previously collected data.

- The system has finite compute for retraining.

- New data comes at irregular intervals and not in fixed-size batches.

- New species or animals must be detected and added to the set of known classes.

*Autonomous self-checkout systems* (Polacco & Backes, 2018; Wankhede et al., 2018) such as Amazon Go use computer vision along with a variety of sensors to detect which items customers have selected, enabling the purchasing of products without the need for a cashier. Below we've listed the relevant aspects of autonomous self-checkout as they pertain to FLUID.

- New products and additional training examples are added to the system over time.

- Images of products used for pretraining differ from real-world data. Each store presents a distribution shift in the visual background, product placement, and lighting (Polacco & Backes, 2018).

- Products in the tail of the distribution have less available real-world examples making classification of rare items more difficult (Brynjolfsson et al., 2006).

- The vision system must recognize which objects held by customers are store products and which are not (out-of-distribution detection).

- Compute for retraining the models is finite.

*Visual defect detection* is the process of identifying defects or imperfections in images or videos. It is commonly used in manufacturing to ensure that products meet specific standards. Computer vision models identify issues including scratches, cracks, dents, or other types of damage on surfaces, as well as defects in the alignment or positioning of components (Czimmermann et al., 2020). We've listed the key details of defect detection that relate to FLUID.

- Models are pretrained on open-source data sets then fine-tuned to fit the client data which often differs from the pretrain distribution.

- Defect examples are collected over time and used to update the model.

- The distribution of defect types is long-tailed and often examples of defects are rare.

- The system ideally should detect new types of defects even if they are outside the training set.

- Defect examples come at irregular intervals and not in fixed-size batches.

- Compute for retraining the models is finite.

