# OpenReview forum: "FLUID: A Unified Evaluation Framework for Flexible Sequential Data"
_TMLR — Accepted by TMLR_

### Review · Reviewer_8uhi · 2022-12-09

**Summary Of Contributions:**

This paper proposes the FLUID benchmark / evaluation framework that aims to evaluate a combination of several key aspects of learning in practical settings, such as sequentially arriving data, X-shot learning and out-of-distribution detection. While fields such as continual learning, few-shot learning and transfer learning tend to focus on evaluating only one of such aspects, this paper aims to jointly evaluate them at the same time. It is argued that this is important because otherwise there is the risk that methods will be developed that are very good at dealing with one aspect, but fail when simultaneously confronted with other aspects. To support this claim, a comprehensive empirical comparison is performed, and it is reported that methods specifically designed for either continual learning or few-shot learning indeed do not perform well on the FLUID benchmark, as they are outperformed by relatively simple baselines.


**Audience:**

Yes

**Broader Impact Concerns:**

No concerns in this regard.

**Claims And Evidence:**

No

**Requested Changes:**

As indicated above, a critical issue to be addressed is the lack of non-stationarity in the benchmark. Either the specification of the benchmark must be changed such that it includes relevant forms of non-stationarity, or the descriptions / interpretations of the benchmark must be changed. If the authors choose the latter option, I would probably also recommend leaving out the continual learning methods from the comparison, to avoid the risk of misinterpretation.

Other issues that might be good to keep in mind when rewriting the paper:
- One finding of the paper is that meta-training based few-shot methods such as prototypical networks and MAML do not perform well on the proposed FLUID benchmark. But I’m wondering whether this finding is correct / fair. It seems to me that the meta-training for these methods is done on test sequences (5-shot 30-way) very different from the actual sequence that the method is tested on. Is it therefore not expected that these methods won’t perform well? Unless I’m missing something, at the moment the paper is written in a way to suggest this is a lot more surprising than it actually is.
- Another finding claimed by the paper is that when a suitable pre-trained network is available, simple approaches (e.g., an NCM classifier on top of the frozen features) often perform very well. I think the paper demonstrates this in a comprehensive and convincing way, and as such I think this is an important contribution, but the insight that when a suitable pre-trained network is available simple approaches work well is not new, at least not in the continual learning field (e.g., Hayes & Kanan 2020 CVPR-W, https://arxiv.org/abs/1909.01520).

Another note I had is that on p18 the authors mention that they provide “a more than fair comparison”. What exactly is meant by “more than fair”? I guess the authors mean that the comparison is in some way stacked against the method that is found to perform best, but this would benefit from more explanation.


**Strengths And Weaknesses:**

Strengths:
- The paper is written and illustrated in a clear style, and it reads very pleasantly.
- The need for a benchmark combining the objectives of continual, few-shot, transfer and representation learning is well-motivated.
- The proposed benchmark is well-described, and I expect there is a good chance that it will be used and adopted by the research community.
- The reported empirical comparisons seem to be performed to good standards.
- Code is provided, and the code seems to be well-documented and neatly organized.

In many ways I think this paper is suitable for publication in TMLR; I think the proposed benchmark / evaluation framework and accompanying code would be useful resources for the research community.

However, I’m afraid there is a critical issue with this paper that makes it in its current form unsuitable for publication. I think this issue is addressable, but it requires a major revision.

This issue is that the sequence of data in the proposed benchmark does not have any non-stationarity. The benchmark is set up such that the data are always sampled i.i.d. from the same underlying distribution. This is problematic firstly because the paper claims it considers the Non-IID aspect (e.g., Table 1), but even more so because a large focus of the paper is on continual learning and continual learning methods, while a defining characteristic of continual learning is non-stationarity in the training data.

The only change in underlying distribution considered in the benchmark is from the pre-training phase to the sequential learning phase, but evaluation is only on the data distribution from the sequential learning phase, making that this falls under the realm of transfer learning, not continual learning (e.g., see Fig 5 of De Lange et al. 2022 TPAMI, https://doi.org/10.1109/TPAMI.2021.3057446).

Probably related to the lack of non-stationarity, the compared continual learning methods also seem to be implemented in rather artificial ways. EWC and LwF are implemented to prevent forgetting of the pre-training data, but the model is not evaluated on the pre-training data, making it unsurprising that these approaches do not perform well in these experiments. With regards to the replay-based methods (DER and ER-ACE), it is not entirely clear how these methods are implemented (e.g., my understanding is that all data is stored, but this is also the case with offline training; how are these then different?). I was also not able to find how hyperparameter-values (e.g., the regularization strength of EWC) were selected and/or which values were used in the end.

---

> ### Author Response · Authors · 2023-01-04
> **Author Response (1/2)**
>
> Thank you for the thorough and thoughtful review. We have revised the manuscript based on your feedback and hope our responses address your concerns and clarify questions.  As advised by the TMLR guidelines, we will wait to submit the revised version after the third review has been received. Below you will find responses to your questions and concerns. If you have more questions or concerns, we would be happy to discuss further.
>
> **The only change in underlying distribution considered in the benchmark is from the pre-training phase to the sequential learning phase, but evaluation is only on the data distribution from the sequential learning phase, making that this falls under the realm of transfer learning, not continual learning.**
>
> We would like to clarify that the data from the sequential learning phase contains 250 pretrain classes and ~22,000 samples from the pretrain distribution. While the formulation of FLUID is different from standard continual learning definitions, catastrophic forgetting of pretrain classes is a significant challenge we observed in our experiments. Empirically, we found that training the full network performed ~10% worse on pretrained classes compared to freezing the backbone which shows that there is significant performance to be gained by mitigating forgetting. Similarly, in figure 3 c), we observed that full network training for too many epochs on the sequential data increases forgetting and leads to ~3% decrease in accuracy.
>
> We explored continual learning techniques (LwF, EWC, DER, and ER-ACE) to understand how well they could prevent forgetting in the FLUID setting. Our empirical conclusion was that large-scale pretraining changes the types of continual learning (CL) methods which are most effective, and we did not intend to discredit the efficacy of standard CL method. We have changed the language in the experiment section to reflect your appreciated feedback.
>
> **EWC and LwF are implemented to prevent forgetting of the pre-training data, but the model is not evaluated on the pre-training data, making it unsurprising that these approaches do not perform well in these experiments.**
>
> We would like to clarify that the model *is evaluated on pretraining data.* The sequential data contains pretrain 250 classes and ~22,000 samples drawn from the pretraining set. We observed that the continual learning methods do improve performance on the pretrain classes compared to training the full network which shows they reduce catastrophic forgetting. The CL methods performed worse compared to methods which froze the backbone and relearned the classifier (fine-tuning, NCM, and Exemplar Tuning). We have revised the methods and experiments section to clarify this point.
>
> **The paper claims it considers the Non-IID aspect (e.g., Table 1), but the training data is stationary.**
>
> Thank you for the feedback. We have revised the paper to more carefully delineate between non-stationary settings such as continual learning and settings with a single distribution shift such as few-shot learning, transfer learning, and FLUID. We have changed the discussion concerning non-IID data in the related work as well as edited the table to reflect your feedback. Initially we categorized all settings where the pretrain data differs from testing/deployment data to be non-IID, but understand that non-stationary is an important aspect to differentiate.
>
> We agree that FLUID differs from the formal continual learning formulation. Our goal was to model the practical setting where models are pretrained then deployed into a sequential setting with an unknown distribution shift. In this setting, methods to prevent catastrophic forgetting of the pretraining data are still needed which was why we evaluated LwF, EWC, etc. Our empirical conclusion was that this setting changes the type of continual learning (CL) approaches which are effective, and we did intend not to discredit standard CL methods.

---

> > ### Comment · Reviewer_8uhi · 2023-01-10
> > **Part of the author response missing?**
> >
> > Thank you for the response to my review. I will wait with responding in detail until the revised manuscript has been uploaded, but I already wanted to raise that I might only be able to see part of your response. The title of your response is "Author Response (1/2)", which suggests there should be two parts, but I can only see one part. Is this correct, or should there be a second part?

---

> > > ### Author Response · Authors · 2023-01-10
> > > **Part 2 now visible**
> > >
> > > Thanks for letting us know. There was an issue with adding reviewers to the readers list which has hopefully been resolved. Part 2 of our response should be visible now.

---

> > > > ### Comment · Reviewer_8uhi · 2023-01-12
> > > > **Thanks**
> > > >
> > > > Great, thank you. I can now indeed see the second part of your author rebuttal as well.

---

> > > > > ### Comment · Reviewer_8uhi · 2023-01-23
> > > > > **Revised version**
> > > > >
> > > > > Dear authors, in a previous response you mentioned that you were planning to upload a revised version of the paper after the third review had been posted, but it seems no revised version has been posted yet (or at least I don't think I can see one). The reviewers are expected to submit a decision recommendation by next week. For this recommendation I would be happy to take a revised version of the paper into account if you can provide one in the next few days.

---

> > > > > > ### Author Response · Authors · 2023-01-23
> > > > > > **Thanks for Following Up**
> > > > > >
> > > > > > Hi Reviewer 8uhi,
> > > > > >
> > > > > > Thanks for following up. We were waiting on a few experiments to finish that we have been added to the revised version. The new revision has been submitted and we have summarized the changes in a new comment above. We appreciate your feedback and would be happy to hear any further comments or concerns you may have.

---

> ### Author Response · Authors · 2023-01-04
> **Author Response (2/2)**
>
> **With regards to the replay-based methods (DER and ER-ACE), it is not entirely clear how these methods are implemented (e.g., my understanding is that all data is stored, but this is also the case with offline training; how are these then different?)**
>
> DER differs from offline training in the use of knowledge distillation and the storage of previous logits. At each offline training phase we store the logits and use them to compute the KL divergence term in the next offline training phase.  For ER-ACE we apply the loss given in the paper:
>
> $\mathcal{L}\_{\text {ace }}(\mathbf{X}^{b f} \cup \mathbf{X}^{i n})=\mathcal{L}\_{ce}(\mathbf{X}^{bf}, C\_{\text {old}} \cup C\_{\text {curr }}) +\mathcal{L}\_{ce}(\mathbf{X}^{in}, C\_{\text {curr}})$
>
> Where $C_{old}$ contains the 1000 pretraining classes and $C_{curr}$ contains the novel classes which have been seen during streaming. For LwF we swept over $\lambda$ values between $[.1, .2, … 1]$. We found .2 maximized mean per-class and overall accuracy. For EWC we used the default hyper-parameters of .1 for lambda and .9 for alpha. We have added the hyper-parameters for these and ER-ACE and DER to section E of the appendix (Implementation Details).
>
>
> **One finding of the paper is that meta-training based few-shot methods such as prototypical networks and MAML do not perform well on the proposed FLUID benchmark. But I’m wondering whether this finding is correct / fair. It seems to me that the meta-training for these methods is done on test sequences (5-shot 30-way) very different from the actual sequence that the method is tested on.**
>
> We agree that adapting meta-trained methods to a more general setting is not straight forward. We experimented with different ways of adapting the meta-training procedure to better match the testing conditions. For prototypical-networks we tried meta-training while randomly sampling the way number between 2-1000 and the shot number between 1-1000 for both conv-4 and ResNet 18. For both experiments the validation and training loss did not converge and performed significantly worse than fixed shot and way values. We also tried ablating over the way number from 1 to 1000 without sampling, which modified how well prototypical networks performed on the head and tail classes, but did not significantly affect overall accuracy. For the Proto-MAML experiments we randomly sampled the way number between 5 and 50 and classes from the WordNet taxonomy in accordance with the original work. We will add these experiments and details to appendix K.
>
> Our intention was not to discount the efficacy of these few-shot methods, but to demonstrate that there is further progress needed in order to scale these sample-efficient methods to more general settings. We have modified the discussion in the experiments and conclusion section to reflect your feedback.
>
>
> **The insight that when a suitable pre-trained network is available simple approaches work well is not new, at least not in the continual learning field (e.g., Hayes & Kanan 2020 CVPR-W, https://arxiv.org/abs/1909.01520).**
>
> Thanks for pointing out this work. We have added it to the related work section and changed the contributions section to reflect the fact that previous works have shown pretraining enables simple approaches in continual learning settings.
>
> **Another note I had is that on p18 the authors mention that they provide “a more than fair comparison”. What exactly is meant by “more than fair”?**
>
> Thanks for the feedback. We mean that NCM does not use a replay buffer whereas other continual learning methods do. We have clarified this statement on p18.

---

> > ### Comment · Reviewer_8uhi · 2023-01-25
> > **Reply to Author Rebuttal (1/2)**
> >
> > Thanks to the authors for their rebuttal and for uploading a revised version of their submission. Please find some comments from me below.
> >
> >
> > My main concern is still that this paper presents FLUID as a continual learning benchmark, despite its lack of non-stationarity in the training data of the sequential learning phase. (For example, at the bottom of p10 it is claimed that FLUID can be considered a specific instance of the CL formulation, after which several differences are mentioned but not the lack of non-stationarity. But see also section 2.) My impression is that the community considers non-stationarity in the training data a defining characteristic of continual learning (combined with performing well on all distributions seen so far), see for example Van de Ven et al. (2022, Nat Mach Intell; https://doi.org/10.1038/s42256-022-00568-3) or De Lange et al. (2021, TPAMI; https://doi.org/10.1109/TPAMI.2021.3057446). If the authors do not agree with this notion, they should at a minimum discuss it.
> > Importantly, this is not just a matter of definitions/semantics, because as a result of this, continual learning methods are applied on FLUID in an artificial way, as further discussed below.
> >
> >
> > **The data from the sequential learning phase contains 250 pretrain classes and ~22,000 samples from the pretrain distribution**
> >
> > I appreciate that there is overlap between the pretrain distribution and the distribution of the sequential learning phase, but I do not think this fundamentally changes things.
> >
> > Let’s say the pre-training data is from distribution A and the sequential learning phase data is from distribution B. Even if there is overlap between A and B, it is not clear to me why one would expect that optimizing for A+B (as the currently implemented continual learning methods attempt to do) would carry advantages over optimizing for B, when one is interested in the performance on B.
> >
> > I think it is clear that the continual learning methods that are applied to the presented benchmark are not intended/suited for this benchmark. It is OK to still try out these methods on the benchmark, but in that case it is important to be upfront about that they would not be expected to work. The authors should not present failure of continual learning methods on this benchmark as a “limitation”, as currently still is the case. Neither should the authors directly contrast their results with findings from the continual learning literature (e.g., as the authors seem to do at the bottom of p2 when claiming “we observe that the type of continual learning methods that are effective changes when large-scale pretraining is available”).
> >
> > I appreciate that the authors not only evaluate just on B, but also on A* (where A* is the part of A that overlaps with B). But this still does not justify using continual learning methods to optimize for A+B. Perhaps this could justify using continual learning methods to optimize or A*+B? But even then, I think it would be important to clarify that such continual learning methods could at most be expected to perform better on A* (i.e., the pre-train classes), and not on B in its entirety.
> >
> >
> > **[T]he methods are evaluated on data from the pretraining distribution and experimental results show that catastrophic forgetting is a challenge in this setting. […] Our goal was to model the practical setting where models are pretrained then deployed into a sequential setting with an unknown distribution shift. In this setting, methods to prevent catastrophic forgetting of the pretraining data are still needed […] Empirically, we found that training the full network performed ~10% worse on pretrained classes compared to freezing the backbone which shows that there is significant performance to be gained by mitigating forgetting.**
> >
> > I think this is an interesting observation, which warrants further investigation. But I do not agree with the interpretation that this must be due to “catastrophic forgetting”. Unless I’m mistaken, this claim also does not seem to be made in the paper itself, only in the rebuttal? Could the authors elaborate on why they think this must be due to catastrophic forgetting? Could this not reflect for example the possibility that optimizing a model for distribution A gives a higher performance on subset A* than optimizing the same model for distribution B does? (where A* is the part of A that overlaps with B) Or that optimizing for distribution B in an online fashion is more challenging than optimizing for distribution B with joint access to all data?

---

> > > ### Comment · Reviewer_8uhi · 2023-01-25
> > > **Replay to Author Rebuttal (2/2)**
> > >
> > > **For EWC we used the default hyper-parameters of .1 for lambda and .9 for alpha**
> > >
> > > What is the “alpha” hyperparameter in EWC? I’m not familiar with that one.
> > >
> > >
> > > **We mean that NCM does not use a replay buffer whereas other continual learning methods do. We have clarified this statement on p18.**
> > >
> > > This does not seem to be clarified, it still says “more than fair comparison”. This is especially problematic in the light of the above discussion that using a replay buffer here (to optimize A+B rather than B, see above), does not seem to be an advantage.
> > >
> > >
> > > **Appendix Q on Non-Stationary Distribution**
> > >
> > > Although I appreciate the effort from the authors in this regard, I don’t think this experiment in the appendix, as currently presented, adds much value, while I think there is a risk of misinterpretation given its succinct description. It is not explicitly stated, but I expect that for this experiment the continual learning methods are implemented in a similar way as before (i.e., trying to prevent forgetting on the initial pre-training data). So even though now there is non-stationarity in the sequential training data, the compared continual learning methods still seem not to be implemented to address that non-stationarity.

---

> > > > ### Author Response · Authors · 2023-01-26
> > > > **Author Response to Reviewer Reply (1/2)**
> > > >
> > > > Thanks for the further discussion and would be happy to discuss any points further. Below you will find our responses and we will upload a revision to reflect your current feedback.
> > > >
> > > > **I appreciate that the authors not only evaluate just on B, but also on A\* (where A\* is the part of A that overlaps with B). But this still does not justify using continual learning methods to optimize for A+B. Perhaps this could justify using continual learning methods to optimize or A\*+B? But even then, I think it would be important to clarify that such continual learning methods could at most be expected to perform better on A\* (i.e., the pre-train classes), and not on B in its entirety.**
> > > >
> > > > To clarify, CL methods which improve performance on A\* could perform better on B given that they do not degrade performance on B\* (the new classes which do not overlap with A). i.e. accuracy of B = accuracy of A\* + accuracy of B\*.
> > > > In practice, we observe that regularization techniques which prevent forgetting (EWC, freezing the backbone)
> > > > *do improve performance* compared to standard training on distribution B *and* B\* (novel classes). In the case of fine-tuning, freezing the backbone leads to ~ 4\% improvement in overall accuracy compared to standard training. If the data set size of B were equivalent to A then the above reasoning that the model could not perform better on B\* would likely be true, but the pretraining data set is significantly larger (~1.2 million versus ~90k). The hypothesis is that the methods prevent forgetting of the large-scale pretraining features which perform better even for novel classes which we see for freezing the backbone (fine-tuning) and EWC. This is discussed further below and in section 5.6 of the paper.
> > > >
> > > > **I do not agree with the interpretation that this must be due to “catastrophic forgetting”. Unless I’m mistaken, this claim also does not seem to be made in the paper itself, only in the rebuttal? Could the authors elaborate on why they think this must be due to catastrophic forgetting?**
> > > >
> > > > The definition of catastrophic forgetting [1] as we understand it is: The weights learned from task A are changed to meet the objectives of a new task, B, causing a decrease in performance on task A. When comparing the frozen backbone to standard training, we see that constraining the weights to that of task A (pretraining) improves performance on A\* (the overlap of A and B) (Table 3). Similarly we observe that CL methods which constrain the change in weights also perform better on A\* (pretrain classes).
> > > > Another result which points to catastrophic forgetting is the experiment in figure 3 c) and section 5.6. Here we observe that there is an optimal number of training epochs for standard training. As the model is trained for longer, the weights move further and eventually the model forgets the pretrain classes (Table 3). We also observe that it performs worse in overall accuracy.
> > > >
> > > > **Could this not reflect for example the possibility that optimizing a model for distribution A gives a higher performance on subset A\* than optimizing the same model for distribution B does? (where A\* is the part of A that overlaps with B)**
> > > >
> > > > We are unclear on this point. We agree that optimizing for distribution A would likely lead to higher accuracy on A\* (pretrain classes), but all of the methods are trained strictly on data from distribution B after the pretraining phase.
> > > > If you mean that methods which regularize the change in parameters (EWC, LwF, freezing the backbone, etc.) would likely perform better on A\* because they do not directly optimize for the training loss with respect to B, then we agree. Preventing accuracy loss on A\* (pretrain classes) while training on B is by definition preventing forgetting so we may misunderstand this point.
> > > >
> > > > The challenge in FLUID is to prevent forgetting of A\* (pretrain classes) while learning from new examples and classes. In table 3, we observe that the CL methods do prevent forgetting of A* compared to standard training with minimal trade-off on learning new classes. For EWC we see higher mean class and overall accuracy compared to standard training, showing that regularization to prevent forgetting does improve overall performance.

---

> > > > ### Author Response · Authors · 2023-01-26
> > > > **Author Response to Reviewer Reply (2/2)**
> > > >
> > > > **Or that optimizing for distribution B in an online fashion is more challenging than optimizing for distribution B with joint access to all data?**
> > > >
> > > > We are unclear on this point as the models do not have joint access to all data. We do agree that optimizing in an online fashion leads to worse features compared to jointly optimizing over a larger data set drawn from B.
> > > >
> > > >
> > > > Along this direction we observe that optimizing over the larger pretraining set in an offline setting likely provides better features, which is why methods for preventing the forgetting of these parameters perform better even on the novel classes (B*) than directly optimizing (standard training) for distribution B. Further evidence for this hypothesis can be found in the update strategies (section 5.6) where we observe that there is an optimal number of training epochs on distribution B which balances the strong pretraining features with learning new classes from distribution B. Training for too long on the smaller, online data set drawn from distribution B leads to lower accuracy for all classes.
> > > >
> > > > **The authors [should not] directly contrast their results with findings from the continual learning literature (e.g., as the authors seem to do at the bottom of p2 when claiming “we observe that the type of continual learning methods that are effective changes when large-scale pretraining is available”).**
> > > >
> > > > We will submit a new revision to convey that CL methods cannot be expected to perform well as the experimental setting is sufficiently different from traditional CL settings. However, we believe it is still important to include that catastrophic forgetting is a significant challenge in the FLUID setting as discussed above, and that there is significant progress to be made in solving it. We also believe that comparing and contrasting approaches such as freezing the parameters to existing CL methods is important as a starting point for new methods which prevent catastrophic forgetting in this setting.
> > > >
> > > > **My main concern is still that this paper presents FLUID as a continual learning benchmark, despite its lack of non-stationarity in the training data of the sequential learning phase. (For example, at the bottom of p10 it is claimed that FLUID can be considered a specific instance of the CL formulation, after which several differences are mentioned but not the lack of non-stationarity.**
> > > >
> > > > We will add discussion of the non-stationarity to p10 and change the content to make it clear that CL methods should not be expected to perform well in this setting.
> > > >
> > > > We believe by the definition presented by Van de Ven et al. (https://www.nature.com/articles/s42256-022-00568-3) that FLUID could be considered task-incremental learning, though we can remove discussion of FLUID as an instance of CL if this is not the case. Van de Ven et al. states: "The third continual learning scenario is ‘class-incremental learning’ (or Class-IL). This scenario is best described as the case where an algorithm must incrementally learn to discriminate between a growing number of objects or classes" (p 1187). From this perspective FLUID would be viewed as class-incremental with 2 episodes where the first episode consists of large-scale pretraining, and the second episode consists of deployment into a distribution with an unknown number of overlapping old and new classes. In traditional class incremental settings, new classes are added at the beginning of each episode. In FLUID, new classes are added throughout the episode as they are drawn from the unknown distribution.
> > > >
> > > > **What is the “alpha” hyperparameter in EWC? I’m not familiar with that one.**
> > > > The alpha referenced on p20 is meant to indicate the coefficient for the task loss.
> > > >
> > > > $$\mathcal{L}(\theta)=\alpha\mathcal{L}_B(\theta)+\sum_i \frac{\lambda}{2} F_i(\theta_i-\theta_\text{A, i}^*)^2$$
> > > >
> > > > The original paper implicitly sets this value to 1. We realize that this was not clear in the appendix, and the new revision has clarified this.
> > > >
> > > > **This does not seem to be clarified, it still says “more than fair comparison”. This is especially problematic in the light of the above discussion that using a replay buffer here (to optimize A+B rather than B, see above), does not seem to be an advantage.**
> > > >
> > > >
> > > > We apologize for missing this edit; we’ve fixed the text on page 18 and will reupload a revision with the suggested changes. We think there may be a misunderstanding about the replay buffer. The replay buffer is for storing training data from distribution B; NCM stores no data during the deployment phase. Using a replay buffer to store data from distribution B is a significant advantage as without it, the models would be forced to discard every sample encountered during the deployment phase and be unable to perform offline training.

---

> > > > > ### Comment · Reviewer_8uhi · 2023-01-27
> > > > > **Reviewer reply**
> > > > >
> > > > > Thank you for your quick response, that is very helpful.
> > > > >
> > > > > I now accept that it can be interesting to test continual learning methods on this benchmark in the way the paper does. Although I think it would make more sense to implement the continual learning methods such that they attempt to prevent forgetting of A* (the part of the pretrain distribution that overlaps with the distribution of the sequential training phase) rather than A (the entire pretrain distribution). I think changing this could improve the paper, but I don’t consider it necessary.
> > > > >
> > > > > I would like to encourage the authors to better motivate in the paper why (and to what extent) continual learning methods might be useful on their benchmark. In particular, I think the hypothesis “the optimal amount of training balances the features learned from ImageNet-1K with those from the smaller, imbalanced streaming data” (as stated in section 5.6) is well-formulated. I recommend the authors to use this hypothesis (or some form of it) to motivate the use of continual learning methods in the first place.
> > > > >
> > > > > Note that the above does not mean that I consider the proposed benchmark to be “continual learning” (as this term is typically used in the literature). The proposed benchmark has a distribution shift (i.e., from pretrain distribution A to sequential training distribution B), but the key difference, which I think the authors still do not discuss, is that in continual learning the goal would be to do well on both A and B, while in the proposed benchmark the goal is to do well on B.
> > > > >
> > > > > Nevertheless, despite that the goal of continual learning methods (i.e., optimize for A+B) does not align with the goal of the proposed benchmark (i.e., optimize for B), I now accept that it can be argued that continual learning methods might be beneficial for this benchmark.
> > > > >
> > > > >
> > > > >
> > > > > **We think there may be a misunderstanding about the replay buffer. The replay buffer is for storing training data from distribution B; NCM stores no data during the deployment phase.**
> > > > >
> > > > > This indeed seems to be something I misunderstood. But this raises new questions. If I now understand it correctly, this means that there is a fundamental difference in implementation between on the one hand EWC and LwF, and on the other hand the replay methods (DER and ER-ACE). My understanding is that EWC and LwF both try to prevent forgetting of the pretraining distribution (i.e., distribution A), while the replay methods try to prevent forgetting of previously seen data from the sequential training data (i.e., distribution B). Is my new understanding correct? If so, could the authors comment on why there is this difference? Either way, it would be good to clarify the description of the way these methods are implemented.

---

> > > > > > ### Author Response · Authors · 2023-01-28
> > > > > > **Reply to Reviewer**
> > > > > >
> > > > > > We appreciate the additional feedback. Below we’ve included our responses and submitted a revision to reflect our recent discussion.
> > > > > >
> > > > > >
> > > > > > **I would like to encourage the authors to better motivate in the paper why (and to what extent) continual learning methods might be useful on their benchmark.**
> > > > > >
> > > > > > We completely agree and have revised the continual learning section (5.2) to reflect the discussion we’ve had on the evidence for the catastrophic forgetting in FLUID.
> > > > > >
> > > > > >
> > > > > > **Note that the above does not mean that I consider the proposed benchmark to be “continual learning” (as this term is typically used in the literature). The proposed benchmark has a distribution shift (i.e., from pretrain distribution A to sequential training distribution B), but the key difference, which I think the authors still do not discuss, is that in continual learning the goal would be to do well on both A and B, while in the proposed benchmark the goal is to do well on B.**
> > > > > >
> > > > > > Valid points and we do not feel strongly whether FLUID falls under the formal definition of continual learning. We included the paragraph about FLUID being considered a specific instance of continual learning under the general formulation given by Prabhu et al. as we did not want to discount previous CL formulations.
> > > > > >
> > > > > > From Prabhu et al.:​​”A realistic formulation of continual learning for classification would be where there is a stream of training samples or data accessible to a learner, each sample comprising a two-tuple $(x_t, y_t)$, where t represents the timestamp or the sample index. Under this setting, at any given $t$, the objective is to provide a mapping $f_{θ_t} : x → y$ that can accurately map a sample $x$ to a label $y ∈ Y_t$.”
> > > > > >
> > > > > > If you feel that considering FLUID as an instance of CL is misleading or incorrect we can remove this section.
> > > > > >
> > > > > > **My understanding is that EWC and LwF both try to prevent forgetting of the pretraining distribution (i.e., distribution A), while the replay methods try to prevent forgetting of previously seen data from the sequential training data (i.e., distribution B). Is my new understanding correct? If so, could the authors comment on why there is this difference?**
> > > > > >
> > > > > > Yes, that is correct. We tried both types of methods (replay and regularization) to see which type of approach was more effective. For the replay methods (DER, ER-ACE), we constrained the difference in output space with respect to a changing model (the model from the previous offline training phase) whereas EWC and LwF are anchored in weight space to the pretrain initialization. This difference in implementation was based on their original formulation and intent and we tried to faithfully adapt them to the FLUID evaluation.

---

> > > > > > > ### Comment · Reviewer_8uhi · 2023-02-03
> > > > > > > **Reviewer reply**
> > > > > > >
> > > > > > > Thank you for the continued discussion.
> > > > > > >
> > > > > > > The definition of continual learning based on Prabhu et al. (2020, ECCV) is very general. It seems to me that this includes the entire online learning literature. It is fine to mention in the paper that according to this definition your proposed benchmark falls under continual learning, but you should also discuss that based on other formulations of continual learning it does not (e.g., based on Van de Ven et al. (2022, Nat Mach Intell) or De Lange et al (2021, TPAMI)). Relatedly, I think it is important to discuss that the objective of continual learning methods (do well on both A and B) does not align with the objective of FLUID (do well on B).
> > > > > > > I think this should be mentioned in the paper, although it is OK to discuss at the same time that despite this mismatch of objectives, continual learning methods might still be useful.

---

> > > > > > > > ### Author Response · Authors · 2023-02-07
> > > > > > > > **Reply to Reviewer**
> > > > > > > >
> > > > > > > > **You should also discuss that based on other formulations of continual learning it does not. Relatedly, I think it is important to discuss that the objective of continual learning methods (do well on both A and B) does not align with the objective of FLUID (do well on B).**
> > > > > > > >
> > > > > > > > You make a good point and we appreciate the feedback. We’ve added discussion of this point along with references to Van de Ven et al. (2022, Nat Mach Intell) and De Lange et al (2021, TPAMI) to the last paragraph of related works and edited the last paragraph of the experiments to reflect these points in our revision.

---

### Review · Reviewer_jwLE · 2022-12-12

**Summary Of Contributions:**

This work proposes a new holistic evaluation (FLUID) for continual learning systems.
Some aspects of the fields of few-shot and open-world are introduced in the framework.
The evaluation is compute-aware, an important consideration for continual-learning systems.

A thorough empirical study demonstrates the effectiveness of new baselines for FLUID over approaches proposed for continual, few-shot and representation learning.

**Audience:**

Yes

**Claims And Evidence:**

Yes

**Requested Changes:**

Properly contrast FLUID with OSAKA, as mentioned above.

Some minor suggestions that could improve the manuscript.
- link Minimum DIstance Thresholding (MDT) with non-parametric expansion methods like, e.g., Chinese-restaurant processes.
- "From the experiments, we find that initializing networks with standard pretraining mitigates the effect of catastrophic forgetting in its current formulation, indicating the need for new evaluation frameworks." Although it is reasonable to pretrain representations with ImageNet when faced with a continual vision problem, more practical learning scenarios might not enjoy such a broad pretaining dataset. It is thus unclear if not incurring forgetting after ImageNet pretraining is an argument for needing new evaluation frameworks.

**Strengths And Weaknesses:**

**Strengths**

Canonical continual-learning evaluation is, in my opinion, far from practical. FLUID is a sensible step towards a more realistic continual-learning evaluation. The empirical study is extensive and offers some insights.

**Weakness**

My main criticism is that FLUID is quite close to a continual-learning scenario proposed over two years ago, namely OSAKA [1]. I understand that staying informed of all related literature can be challenging. This one, however, should definitely be referenced and properly contrasted. Listed below are OSAKA's properties:
- Open-World = yes. (the set of known classes may change over time)
- Sequential = yes.
- variable batch-size = N/A (i.e. not a requirement)
- few-shot and many-shot = yes
    - OSAKA is essentially a continual few-shot learning problem, such that the systems will be tested few-shot the first time a data distribution appears and many-shots later.
- compute-aware = No.
- memory constrained = N/A.
- Flexible training = Yes.
   - in OSAKA, systems sporadically encounter new data distribution and have to autonomously discern if the data is OoD and requires retraining.
- non-IID: Yes.

Both settings offer a pertaining phase and a continual learning phase that includes new data distributions.
Both settings feed the data online to the learners and only reveal the labels after the prediction.
In short, FLUID adds the aspect of compute awareness to the OSAKA's evaluation.
Noteworthily, OSAKA is more flexible as $p(y|x)$ is allowed to shift during the "life/deployment" of the systems.

[1] Online fast adaptation and knowledge accumulation (OSAKA): a new approach to continual learning

---

> ### Author Response · Authors · 2023-01-04
> **Author Response**
>
> Thank you for the thorough and thoughtful review. We have revised the manuscript based on your feedback and hope our responses address your concerns and clarify questions. As advised by the TMLR guidelines, we will wait to submit the revised version after the third review has been received. Below you will find responses to your questions and concerns. If you have more questions or concerns, we would be happy to discuss further.
>
> **Compare and contrast with OSAKA [1].**
>
> Thanks for pointing us to this interesting work. We agree that the frameworks are very similar in formulation and objective. Below we've contrasted FLUID and OSAKA.
>
> Formulation Differences
> - FLUID considers the trade-off between compute and accuracy. Flexible training phases enable diverse training strategies such as meta-training and replay buffers which can differ greatly in the compute cost. For comparison and integration of methods and training strategies, it is important to consider both dimensions.
>
> Empirical Differences
> - The FLUID experiments are conducted at a significantly larger scale. OSAKA evaluates on primarily few-shot datasets such as Omniglot, MNIST, and Tiered-ImageNet. While larger-scale is not inherently better, we found that larger datasets can change empirical conclusions such as meta-trained networks not scaling to the FLUID subset of ImageNet-22k.
>
> - FLUID samples the sequential data from a long-tailed setting and measures the accuracy on tail/head classes. This allows us to study the trade-off in accuracy between head/tail classes. For example, we observe that few-shot techniques improve performance on tail classes, but sacrifice performance for head classes.
>
> We have added the comparison of OSAKA to FLUID to the related works, and added it to the comparison chart in Table 1.
>
> **Link Minimum DIstance Thresholding (MDT) with non-parametric expansion methods like, e.g., Chinese-restaurant processes.**
>
> Thanks for the feedback. Like you suggested, similar to the DP-Means algorithm [2], Minimum Distance Thresholding with a Nearest Class Mean classifier can be derived from a Dirichlet process mixture. In this model we consider a sample to be out-of-distribution if it is assigned to a new cluster. The concentration parameter for the Chinese Restaurant process can be related to the out-of-distribution threshold, $\tau$, as:
>
> $\tau=2 \sigma \log \left(\frac{\alpha}{\left(1+\frac{\rho}{\sigma}\right)^{d / 2}}\right)$
>
> where $\rho$ is the covariance scaling of the gaussian prior over the cluster means ($\boldsymbol{\mu} \sim \mathcal{N}(\mathbf{0}, \rho I)$) and $\sigma$ is the scaling for the isotropic covariance of each gaussian cluster according to $\mathcal{N}(\boldsymbol{\mu}_{\boldsymbol{z}_i}, \sigma I)$.
>
> Similar to the DP-Means derivation, as $\sigma$ goes to 0, the probability of a sample $x_i$ being assigned to a new cluster goes to 1 when the distance of $x_i$ to the closest cluster, $\mu\_{min}$, exceeds $\tau$.
>
> We have added discussion of the connection to non-parametric methods to the methods section and appendix.
>
> **From the experiments, we find that initializing networks with standard pretraining mitigates the effect of catastrophic forgetting in its current formulation, indicating the need for new evaluation frameworks." Although it is reasonable to pretrain representations with ImageNet when faced with a continual vision problem, more practical learning scenarios might not enjoy such a broad pretaining dataset. It is thus unclear if not incurring forgetting after ImageNet pretraining is an argument for needing new evaluation frameworks.**
>
> Thanks for the feedback. We agree some scenarios may not have the benefit of large-scale pretraining and forgetting even with pretraining is still a challenge. Our intent was to convey that large-scale pretraining changes the types of methods which are effective for preventing forgetting. For such scenarios, new methods developed under such conditions can outperform standard continual learning approaches as observed in our experiments. We have revised the discussion in the experiments and conclusion to better convey this point.
>
>
>
> [1] Online fast adaptation and knowledge accumulation (OSAKA): a new approach to continual learning
>
> [2] Revisiting k-means: New algorithms via Bayesian nonparametrics.

---

### Review · Reviewer_B1ZX · 2023-01-16

**Summary Of Contributions:**

The paper develops a new evaluation framework for learning settings that involve processing sequentially arriving data that enables comparison of methods, including few-shot learning and continual learning, across several learning settings in this area. Extensive experimental exploration is offered to study the shortcomings of existing methods and then research questions are defined for future exploration.

**Audience:**

Yes

**Claims And Evidence:**

Yes

**Requested Changes:**

I believe I have not reviewed a manuscript similar to this work that lacks any algorithmic novelty and is not introducing a new dataset. I find it challenging how this work can be improved. I have the following comments which may help:

1. Problem motivation needs to be clarified further. I am not convinced that there is a need to have a unified evaluation framework for the learning settings described in the paper. It is claimed, "in many applications the exact conditions cannot be known a priori and are likely to change over time". I tend to disagree with this statement because the user likely can predict which setting is more appropriate. Could you give some examples of those "many applications"? I like to see a clear motivation with connections to real-world applications for this work.

2. Catastrophic forgetting is a major concern in CL. I would like to see more comparisons and justification why the existing benchmarks are not sufficient for this purpose.

3. I would like to see algorithmic novelty, even at a simple level. I have difficulty convincing myself that this work is suitable for a venue such as TLMR. I understand empirical explorations are extensive but they lack to convey a clear justification for this work.

I am open for further discussion about the contributions of this work.

**Strengths And Weaknesses:**

Strengths:

1. The paper reads well and is easy to follow.

2. Empirical exploration includes many methods from the studied learning settings and offers comparisons that previously have not been made.

Weaknesses:

1. The motivation is weak. It is unclear why such an evaluation may be beneficial in practice.

2. The algorithmic novelty in the proposed framework is limited and incremental. I am not sure if this level of novelty is sufficient for TMLR. This work may be suitable for a dataset and benchmark venue.

3. The overall empirical contribution is not very novel as existing datasets are used and only the evaluation framework is novel.

---

> ### Author Response · Authors · 2023-01-19
> **Author Response (1/2)**
>
> Thank you for the thorough review. We hope our responses address your concerns and questions. If you have further questions or concerns, we would be happy to discuss further. We will submit a revision of the paper soon based on your and the other reviewers' feedback.
>
>  **The motivation is weak. It is unclear why such an evaluation may be beneficial in practice. I am not convinced that there is a need to have a unified evaluation framework for the learning settings described in the paper. Could you give some examples of those "many applications"? I like to see a clear motivation with connections to real-world applications for this work.**
>
> Below we’ve detailed real-world applications that share most, if not all, the key aspects of FLUID. The goal of FLUID is to foster research towards methods which are robust to the adverse learning conditions encountered in real-world applications.
>
> *Computer vision for monitoring biodiversity* [1][3]. Researchers use camera traps to identify and track animal populations in order to monitor ecological systems. Beery et al. [1] discuss the challenges of deployment which are similar to those in the FLUID evaluation. Specifically, they cite difficulty in generalizing to new environments, lack of examples for rare classes, and lengthy, intermittent data collection. We’ve listed the relevant details of monitoring biodiversity as they pertain to FLUID.
>
> - The model is pretrained on wildlife data which differs from the deployment distribution.
> - The camera trap systems are deployed and sequentially collect data from the new environment.
> - As with many real-world datasets, species observation data exhibits a long-tail distribution [1][2]. Rare species must be classified using only a few examples.
> - The system may be queried and updated at any time using previously collected data.
> - The system has finite compute for retraining.
> - New data comes at irregular intervals and not in fixed-size batches.
> - New species or animals must be detected and added to the set of known classes.
>
> [1] Beery, Sara, et al. "Recognition in terra incognita." ECCV (2018)
>
> [2] Sudderth, Erik, and Michael Jordan. "Shared segmentation of natural scenes using dependent Pitman-Yor processes." NeurIPS (2009)
>
> [3] Tuia, Devis, et al. "Perspectives in machine learning for wildlife conservation." Nature communications (2022)
>
> *Autonomous self-checkout systems* [4][5] such as Amazon Go use computer vision along with a variety of sensors to detect which items customers have selected, enabling the purchasing of products without the need for a cashier. Below we’ve listed the relevant aspects of autonomous self-checkout as they pertain to FLUID.
>
> - New products and additional training examples are added to the system over time.
> - Images of products used for pretraining differ from real-world data. Each store presents a distribution shift in the visual background, product placement, and lighting [5].
> - Products in the tail of the distribution have less available real-world examples making classification of rare items more difficult [6].
> - The vision system must recognize which objects held by customers are store products and which are not (out-of-distribution detection).
> - Compute for retraining the models is finite.
>
> [4] Polacco, Alex et al.. "The amazon go concept: Implications, applications, and sustainability." Journal of Business and Management
>
> [5] Wankhede, Kirt et al.. "Just walk-out technology and its challenges: A case of Amazon Go." ICIRCA (2018)
>
> [6] Brynjolfsson, Erik et al.. "From niches to riches: Anatomy of the long tail." Sloan management review
>
> *Visual defect detection* is the process of identifying defects or imperfections in images or videos. It is commonly used in manufacturing to ensure that products meet specific standards. Computer vision models identify issues including scratches, cracks, dents, or other types of damage on surfaces, as well as defects in the alignment or positioning of components [7]. We’ve listed the key details of defect detection that relate to FLUID.
>
> - Models are pretrained on open-source data sets then fine-tuned to fit the client data which often differs from the pretrain distribution.
> - Defect examples are collected over time and used to update the model.
> - The distribution of defect types is long-tailed and often examples of defects are rare [8].
> - The system ideally should detect new types of defects even if they are outside the training set.
> - Defect examples come at irregular intervals and not in fixed-size batches.
> - Compute for retraining the models is finite.
>
> [7] Czimmermann, Tamás, et al. "Visual-based defect detection and classification approaches for industrial applications—a survey." Sensors (2020)
>
> [8] X. He and X. Qian, "A real-time surface defect detection system for industrial products with long-tailed distribution," Industrial Electronics and Applications Conference (2021)

---

> ### Author Response · Authors · 2023-01-19
> **Author Response (2/2)**
>
> **Catastrophic forgetting is a major concern in CL. I would like to see more comparisons and justification why the existing benchmarks are not sufficient for this purpose.**
>
> We agree that catastrophic forgetting is still a major challenge to be solved. Recent works [10][11][12] including FLUID show that progress on existing benchmarks does not necessarily generalize to more realistic settings. In our experiments we show strengths and weaknesses of current ML techniques when conditions differ from the standard, controlled benchmarks. For example, the type of CL methods which perform well differ when starting from a pretrained model.
>
> In addition to our own experiments, we’ve summarized three other works which empirically demonstrate shortcomings of existing continual learning benchmarks. In particular, [12] highlights the significant gap between academic benchmarks and real-world applications of CL.
>
> - Prabhu et al. [10] show in their work (ECCV Oral) that flawed assumptions such as not measuring compute in standard continual learning benchmarks allow for a naive baseline to drastically outperform state-of-the-art methods.
> - Hussain et al. [11] show that catastrophic forgetting in the general lifelong learning benchmark can be largely attributed to the experimental design rather than modeling limitations. Furthermore, they show that correcting for the limitations leads to a simple replay baseline outperforming state-of-the-art memory-based methods.
> - A recent paper under review at ICLR [12] discusses the gap between academic benchmarks and real-world industry applications. In particular, they point out the unrealistically small replay buffer and artificial task boundaries. Similar to our experiments, they empirically find that EWC [13] and MIR [14] perform worse than a simple baseline in a more general setting.
>
>
>
> [10] Prabhu, Ameya et al. "Gdumb: A simple approach that questions our progress in continual learning." ECCV (2020)
>
> [11] Hussain, Aman, et al. "Towards a robust experimental framework and benchmark for lifelong language learning." Thirty-fifth Conference on Neural Information Processing Systems Datasets and Benchmarks Track
>
> [12] “Online Continual Learning for Progressive Distribution Shift (OCL-PDS): A Practitioner's Perspective”
>
> [13] Kirkpatrick, James, et al. "Overcoming catastrophic forgetting in neural networks." Proceedings of the national academy of sciences (2017)
>
> [14] Aljundi, Rahaf et al. "Online Continual Learning with Maximally Interfered Retrieval."
>
> **The algorithmic novelty in the proposed framework is limited and incremental. I am not sure if this level of novelty is sufficient for TMLR. This work may be suitable for a dataset and benchmark venue.**
>
> The baselines we propose, Exemplar Tuning and Minimum Distance Thresholding, although simple, outperform existing methods in FLUID. The goal of proposing such baselines is not algorithmic novelty, but to serve as a starting point for further research and to show that there is progress to be made in developing methods for more general settings.
>
> Additionally, we’d like to point out that TMLR (https://jmlr.org/tmlr/acceptance-criteria.html) emphasizes novelty should not be the main criteria for acceptance:
> “Nor should it form the basis for rejecting work on a method considered not “novel enough”, as novelty of the studied method is not a necessary criteria for acceptance. We explicitly avoid these terms (“significant”, “impactful”, “novel”), and focus instead on the notion of “interest”. If the authors make it clear that there is something to be learned by some researchers in their area from their work, then the criteria of interest is considered satisfied.”
>
> TMLR (https://jmlr.org/tmlr/editorial-policies.html#evaluation) includes the following areas of research in their scope of submissions and we believe that the paper fits into these categories:
> - Accounts of applications of existing techniques that shed light on the strengths and weaknesses of the methods.
> - Formalization of new learning tasks (e.g., in the context of new applications) and of methods for assessing performance on those tasks.

---

> > ### Comment · Reviewer_B1ZX · 2023-01-25
> > **Response to feedback**
> >
> > Dear authors,
> >
> > Thank you for your thorough response. based on your response and the rest of the reviews, I think my judgment about the algorithmic novelty of this work is validated. However, you also have a good point that others can learn from your work. What I like to be more clear is to highlight that prior methods are failing on your benchmark and there is a huge room for improvement so that other researchers are motivated to use FLUID in future works.

---

> > > ### Author Response · Authors · 2023-01-28
> > > **Reply to Reviewer**
> > >
> > > Thanks for the additional feedback. You make a great point that the focus should be about the analysis of existing methods and the significant room for progress on FLUID. Based on your feedback we've modified the language in the latest revision to better emphasize the limitations of current methods and the room for progress on FLUID.
> > >
> > > Your points on novelty are completely valid. The intended contribution of FLUID, as you mentioned, is to introduce a setting which more closely modeled real-world applications, and to provide empirical insights about existing methods which show limitations and potential areas for progress.
> > >
> > > If you have any additional feedback or concerns we'd be happy to discuss.

---

### Author Response · Authors · 2023-01-23
**Summary of Changes**

We would like to thank the reviewers for their thorough reviews. In response to their feedback we have made the suggested revisions. We hope that our responses and revision clarify questions and address the concerns of the reviewers. We would be happy to discuss any further questions or concerns. Changes in the revised version are colored with orange text. Below we address overall points then summarize changes by section:

### Overall Changes
Reviewer 8uhi had valid concerns about comparing continual learning with FLUID. They pointed out that methods are not evaluated on pretraining data and the deployment distribution is stationary. To address the first concern, we have clarified in the paper that the methods *are evaluated on data from the pretraining distribution* and experimental results show that catastrophic forgetting is a challenge in this setting. To address the point about non-stationary distributions, we have revised the related works and Table 1 to clarify that the distribution shift in FLUID differs from the non-stationary setting. We have also changed the language throughout the paper to clarify that we are not discounting the efficacy of the CL methods, and our goal is to show that experimental details can change experimental conclusions about which methods are effective. Finally, we’ve added experiments to the appendix (Table 15) which show results for a non-stationary distribution (FLUID is evaluated on the 3 streaming sequences in alternating fashion).



### Introduction
- Reviewer 8uhi pointed pointed us to the work of Hayes & Kanan [1] which had similar empirical findings about the effect of pretraining on CL methods. We have edited the discussion regarding this finding and acknowledged this work.

### Related Work
- Reviewer jwLE pointed us to the relevant work of Caccia et al. [2] which proposes a similar evaluation. We have added a paragraph to compare and contrast FLUID with the OSAKA benchmark and added OSAKA to the comparison table.
- Based on the feedback from reviewer 8uhi we more carefully delineate between non-stationary and distribution shift in the related works and have edited table 1 to reflect this.
- Based on the feedback from reviewer AB1ZX we have added references and discussion about papers which point out the shortcomings of continual learning evaluations.

### Methods
- Based on the feedback from Reviewer jwLE we have detailed how our proposed baseline of Minimum Distance Thresholding can be derived from a Dirichlet Process Mixture model in the Out-of-Distribution (OOD) Methods section.

### Experiments
- We added experiments which explore different schemes for meta-training based on the comments from Reviewer 8uhi to the few-shot learning section and in table 14 of the appendix.
- We added experiments to the appendix (table 15) )which evaluates FLUID in a non-stationary setting where the set of classes in the distribution can change.

### Conclusion
- We changed the language regarding the continual learning methods to make it clear we are not discounting the existing CL methods and simply observing that different types of methods may be needed to prevent forgetting in more general settings.

### Appendix
- Added implementation details and hyper-parameters for continual learning methods to Appendix E based on feedback from Reviewer 8uhi.
- Added table 14 and 15 for the meta-training ablation and non-stationary experiments.

[1] Lifelong Machine Learning with Deep Streaming Linear Discriminant Analysis

[2] Online fast adaptation and knowledge accumulation (OSAKA): a new approach to continual learning

---

### Decision · Action_Editors · 2023-02-15

**Recommendation:** Accept with minor revision

**Comment:**

All reviewers agree that the latest version of the paper meets the criteria for acceptance at TMLR, congratulations!

The discussion period was thorough and I thank the authors for carefully considering the comments of the reviewers and updating their paper several times accordingly. In particular, the added connection to the literature and, even more importantly, the more precise discussion of the differences between the FLUID evaluation and the "standard" CL training/testing better emphasize the contribution of this paper. In my view, the improved clarity is likely to increase the usefulness of this contribution to the community.

Minor revision. You responded to reviewer B1ZX's comments regarding the motivation of the setting with a few interesting potential applications. I encourage you to add (part of) this discussion to the paper (e.g., in passing in the introduction or in a paragraph or two in the related works or the appendix). This will better justify this claim "However, in many applications, the exact conditions cannot be known a priori and are likely to change over time."

**Audience:**

This framework draws ideas from published work in the continual learning literature (including OSAKA). While it proposes a different setting compared to most CL literature it is likely to interest researchers from that community.

FLUID is also likely to be interesting to transfer-learning and few-shot learning audiences.

Ideas of (online) learning under the distribution shift are of broad interest and this work could appeal to researchers wanting to evaluate methods in the FLUID setting and/or extend the framework for more specific use cases.

**Claims And Evidence:**

The reviewers unanimously agree that the claims in the current revision of the paper are supported by accurate, convincing, and clear evidence.

The reviewers were already in broad agreement in their initial reviews. They mentioned that the need for this benchmark was well-motivated, the paper was written clearly, and the empirical comparison was valid and extensive.

The discussion with reviewers and the revisions provided by the authors improved the accuracy of the claims notably as it relates to the connection between FLUID and the existing CL literature.

Based on the reviews, there is one remaining claim that I believe should be justified. I discuss it below in the Comments Section.

---

> ### Author Response · Authors · 2023-03-15
> **Thanks for the Feedback. Camera Ready Revision Submitted.**
>
> Thanks for the feedback on adding the discussion of real-world applications to the paper. We added brief discussion in the introduction and referenced to additional paragraphs in the appendix. We've uploaded the new version in the camera-ready revision. We appreciate the valuable and thoughtful feedback given by all reviewers.
>
> Regards, Authors